# Complex water networks visualized by cryogenic electron microscopy of RNA

Rachael C. Kretsch[1,10], Shanshan Li[2,10], Grigore Pintilie[3,10], Michael Z. Palo[4], David A. Case[5], Rhiju Das[1,6,7 ✉], Kaiming Zhang[2 ✉] & Wah Chiu[1,3,8,9 ✉]

The stability and function of biomolecules are directly influenced by their myriad interactions with water[1–16]. Here we investigated water through cryogenic electron microscopy (cryo-EM) on a highly solvated molecule: the *Tetrahymena* ribozyme. By using segmentation-guided water and ion modelling (SWIM)[17,18], an approach combining resolvability and chemical parameters, we automatically modelled and cross-validated water molecules and $Mg^{2+}$ ions in the ribozyme core, revealing the extensive involvement of water in mediating RNA non-canonical interactions. Unexpectedly, in regions where SWIM does not model ordered water, we observed highly similar densities in both cryo-EM maps. In many of these regions, the cryo-EM densities superimpose with complex water networks predicted by molecular dynamics, supporting their assignment as water and suggesting a biophysical explanation for their elusiveness to conventional atomic coordinate modelling. Our study demonstrates an approach to unveil both rigid and flexible waters that surround biomolecules through cryo-EM map densities, statistical and chemical metrics, and molecular dynamics simulations.

Advances in cryo-EM have enabled the visualization of biomolecular complexes in their near-native hydrated states. Water and ions are critical for maintaining the stability and functional effectiveness of biomolecules[1–3]. Unlike most protein or protein–RNA complexes, RNA forms well-defined structures whose cores are extensively solvated. Water has been implicated in the stability, catalysis and dynamics of RNA both independently and in collaboration with ions[4–11]. Both highly ordered and diffuse water and ions are involved in RNA folding and function[19,20]. Thus, RNA-only structures offer unique opportunities to understand how water interacts with and stabilizes biomolecules. Unfortunately, the flexibility of RNA raises challenges for experimental structure determination. Molecular dynamics simulations can suggest kinetic and structural information inaccessible to current cryo-EM and X-ray crystallographic methods[12]. For example, molecular dynamics studies have proposed binding sites for long-lived water[6,13–15], $Mg^{2+}$ ions[14,21] and spines of fully hydrated metal ions[16]. Nevertheless, the sensitivity and potential inaccuracies of the parameterization and classical assumptions of molecular dynamics force fields have limited confidence in these inferences[22–24].

Here we report cryo-EM of solvated and enzymatically active *Tetrahymena* ribozyme without substrate (387 nt, 128 kDa) using cryo-EM in two independent reconstructions at 2.2 Å and 2.3 Å resolution, enabling detailed analysis and confidence estimation of the interactions of ordered water in the context of an intricate RNA tertiary structure. Water is extensively observed in the interior of the ribozyme and can directly mediate interactions between RNA atoms without also coordinating site-bound ions. Unexpectedly, we found numerous sites where computationally identified waters in the 2.2 Å and 2.3 Å maps disagreed but still had cryo-EM densities that were highly similar. Many of these regions showed a high correlation between the cryo-EM densities and the density predicted by explicit solvent all-atom molecular dynamics simulations, suggesting that cryo-EM maps contain information on highly mobile waters in addition to ordered waters traditionally annotated in atomic models. We also observed regions where the two cryo-EM maps share diffuse water density features that do not agree with the predictions of molecular dynamics, encouraging further development and validation of molecular dynamics force fields, potentially through future blind prediction challenges using flash-frozen biomolecules.

## Structure determination

Since 2018, cryo-EM has been applied to the study of several RNA-only 3D structures at subnanometre resolutions[25], but obtaining sufficient resolution to resolve waters has remained out of reach. Meanwhile, cryo-EM single-particle analysis has been effective at determining atomic structures of large protein–nucleic acid complexes at approximately 2 Å resolution, enabling the visualization of ordered water and ion densities[26,27], but nucleic acids in these complexes are generally not fully solvated because of their extensive interactions with proteins.

[1]Biophysics Program, Stanford University School of Medicine, Stanford, CA, USA. [2]Department of Urology, The First Affiliated Hospital of USTC, MOE Key Laboratory for Cellular Dynamics, Center for Advanced Interdisciplinary Science and Biomedicine of IHM, Division of Life Sciences and Medicine, University of Science and Technology of China, Hefei, China. [3]Department of Bioengineering and James Clark Center, Stanford University School of Medicine, Stanford, CA, USA. [4]Department of Structural Biology, Stanford University School of Medicine, Stanford, CA, USA. [5]Department of Chemistry and Chemical Biology, Rutgers University, Piscataway, NJ, USA. [6]Department of Biochemistry, Stanford University School of Medicine, Stanford, CA, USA. [7]Howard Hughes Medical Institute, Stanford University, Stanford, CA, USA. [8]Department of Microbiology and Immunology, Stanford University School of Medicine, Stanford, CA, USA. [9]Division of CryoEM and Bioimaging, SSRL, SLAC National Accelerator Laboratory, Menlo Park, CA, USA. [10]These authors contributed equally: Rachael C. Kretsch, Shanshan Li, Grigore Pintilie. ✉e-mail: rhiju@stanford.edu; kmzhang@ustc.edu.cn; wahc@stanford.edu

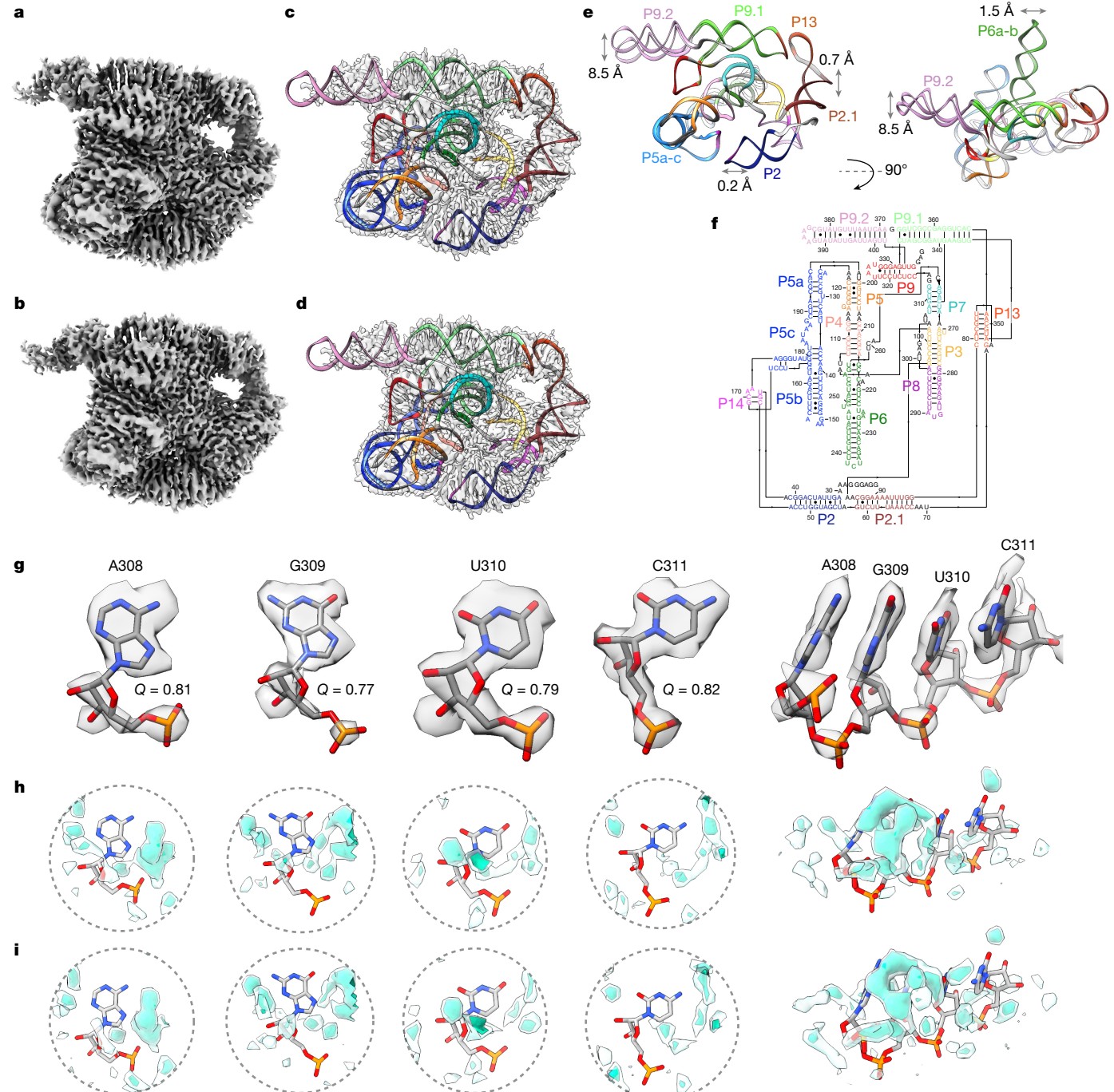

**Fig. 1 | Map and model of *Tetrahymena* ribozyme at 2.2 Å and 2.3 Å resolution.** **a,b**, 2.2 Å (**a**) and 2.3 Å (**b**) cryo-EM map at 3σ threshold. **c,d**, Maps at 2.2 Å (**c**) and 2.3 Å (**d**) with transparent surface and derived model in ribbon display, each domain is coloured. **e**, Models from 2.2 Å and 2.3 Å maps are overlaid. Deviations are labelled for peripheral domains that have considerable differences. **f**, Secondary structure diagram. All domains are labelled and coloured.

**g**, Extracted densities around four nucleotides showing base resolvability and clear separation of stacked bases. The *Q* score is labelled. **h,i**, Density surrounding four nucleotides in the 2.2 Å (**h**) and 2.3 Å (**i**) maps showing similar density features surrounding the RNA. The map was segmented at 3σ using Segger[59]. Density segments at distances 1.8–5.0 Å from the RNA heavy atoms are displayed at 5σ (transparent teal) and 8σ (dark teal).

This study extends the resolution of an RNA-only system, *Tetrahymena* ribozyme[28], towards 2 Å using the same sample preparation for the apo-ribozyme as previously described[29], but with more data and a next-generation electron detector (Extended Data Table 1 and Methods). Single-particle cryo-EM analysis yielded two maps after image classification at 2.2 Å and 2.3 Å resolution with Rosenthal–Henderson B-factor values[30] of 63 Å² and 66 Å², respectively (Fig. 1a,b, Extended Data Fig. 1 and Methods). Overlaying the two models shows that the structural differences lie primarily in peripheral domains P9.2 and P6,

which both point out into solution without interactions with the rest of the ribozyme (Fig. 1c–f and Methods).

## Structure quality assessment

In the cryo-EM maps, unambiguous separation between bases was visually evident and bases were well resolved, indicating that there is high confidence in the positioning of nucleotides (Fig. 1g and Extended Data Fig. 2a,b). When zooming out, most domains in the structure had

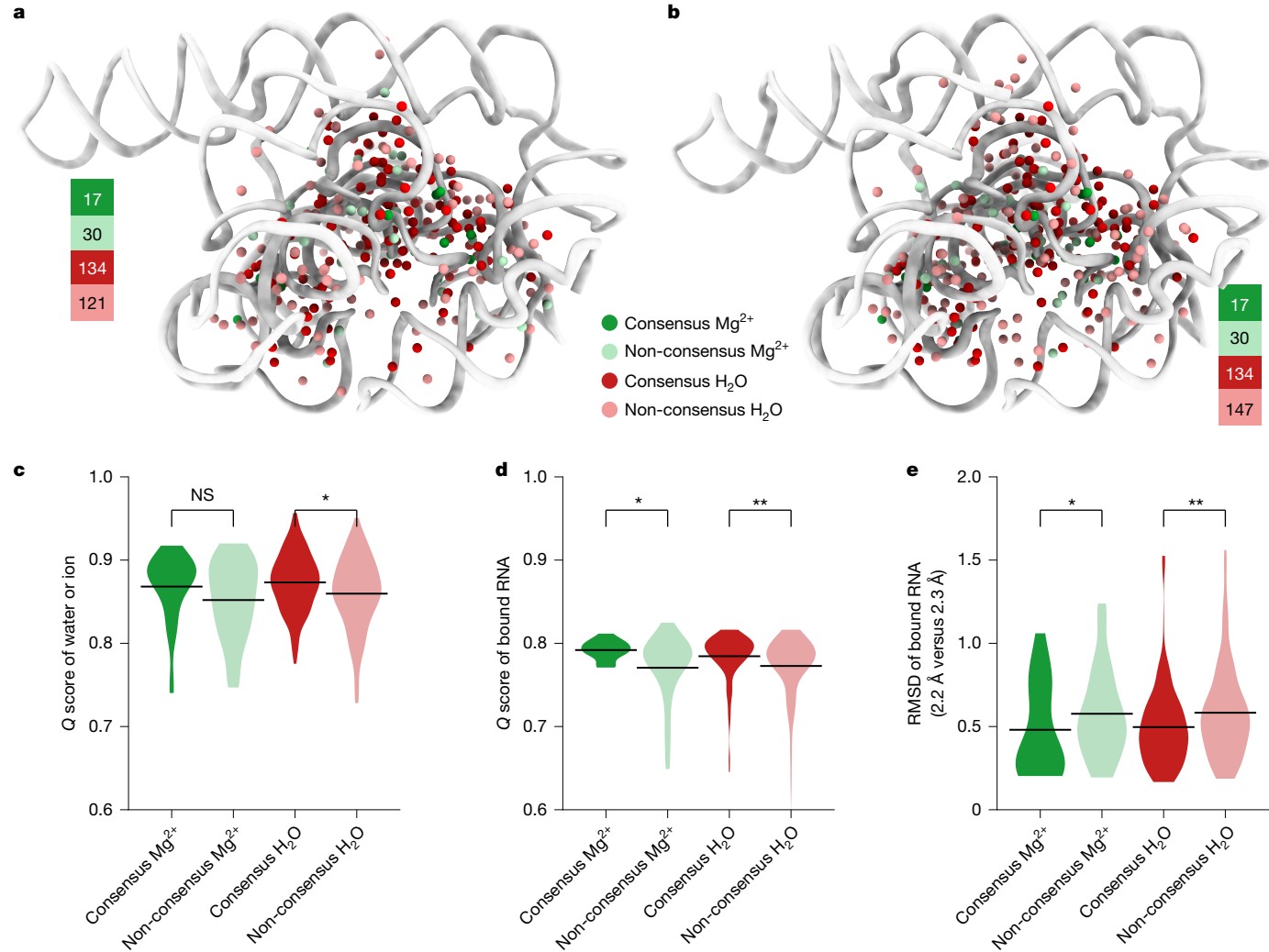

**Fig. 2 | Water and Mg²⁺ ion detection with SWIM and consensus between 2.2 Å and 2.3 Å maps. a,b,** Detected waters (red) and Mg²⁺ ions (green) for the 2.2 Å (**a**) and 2.3 Å (**b**) model. Consensus water and Mg²⁺ ions coloured dark and counts are noted next to each model. **c–e,** Distributions of water and Mg²⁺ ions separated by consensus and non-consensus types: $Q$ score of water or Mg²⁺ ions (**c**; $P_{Mg}$ = 5.7 × 10⁻² and $P_{water}$ = 6.1 × 10⁻⁴), average $Q$ score of bound RNA nucleotides (**d**; $P_{Mg}$ = 2.2 × 10⁻³ and $P_{water}$ = 1.0 × 10⁻⁶) and RMSD of bound RNA nucleotides between the 2.2 Å and 2.3 Å models after alignment on all RNA heavy atoms (**e**; $P_{Mg}$ = 2.6 × 10⁻² and $P_{water}$ = 3.8 × 10⁻⁵). The horizontal line is the mean value. Pairwise significance was determined by a two-sided Mann–Whitney $U$-test: not significant (NS) $P > 0.05$, *$P < 0.05$ and **$P < 10⁻⁴$.

a $Q$ score[31], a metric to assess the resolvability of atoms, above that expected at the overall resolution of the map (Extended Data Fig. 2c). Additional density peaks were observed around the RNA, potentially water, ions or experimental noise (Fig. 1h,i). Although water had been observed previously in X-ray crystal structures of the P4–P6 subdomain of the ribozyme[8,32], only ions had been previously modelled in structures including the catalytic domain of the ribozyme[28,29,33–35]. The assignment of water and ion densities in X-ray diffraction models has been heavily scrutinized[36,37], highlighting the need for rigorous analysis. Rigorous assessment of map quality and modelling of the RNA atoms had to be established to avoid modelling water into noise peaks, RNA density or ion density. As we observed that the cryo-EM density is not uniformly resolved across domains of the ribozyme (Extended Data Fig. 3), water could only be confidently modelled in select regions. These well-resolved regions were also conformationally very similar between the two models (root mean square deviation (RMSD) of 0.59 Å; Extended Data Fig. 3) and thus we reasoned that these regions would be well suited for modelling water with rigorous criteria and comparing these placements between the two maps as cross-validation.

## Automated modelling of water with SWIM

For modelling water, we applied SWIM, which was originally developed for automated analysis of water and ions in atomic-resolution cryo-EM maps of proteins[17,18]. Several SWIM criteria were updated or introduced to be more stringent and reduce the likelihood of modelling water in noise peaks at the 2.2–2.3 Å resolution observed here, resulting in conservative peak identification (Extended Data Fig. 4a–i and Methods). Owing to resolution limitations, resolvability was only sufficient in the solvent shell directly adjacent to the RNA atoms, hence we strictly limited peak assignments to this shell. Using SWIM, 255 and 281 water molecules along with 47 and 47 Mg²⁺ ions were modelled in the 2.2 Å and 2.3 Å ribozyme maps, respectively (Figs. 2a,b and 3). A small number of sites are expected to be partially occupied by monovalent ions, but we expect the contribution to be minor compared with water and Mg²⁺ ions (see Extended Data Figs. 6k–m and 9e,f for detailed discussion). The modelled waters were restricted to those with high resolvability as indicated by the $Q$ scores in the final maps of approximately 0.8, with a trend to higher $Q$ score for waters bound to more RNA atoms (Extended Data Fig. 4j–m). The majority of waters were modelled closest to oxygen

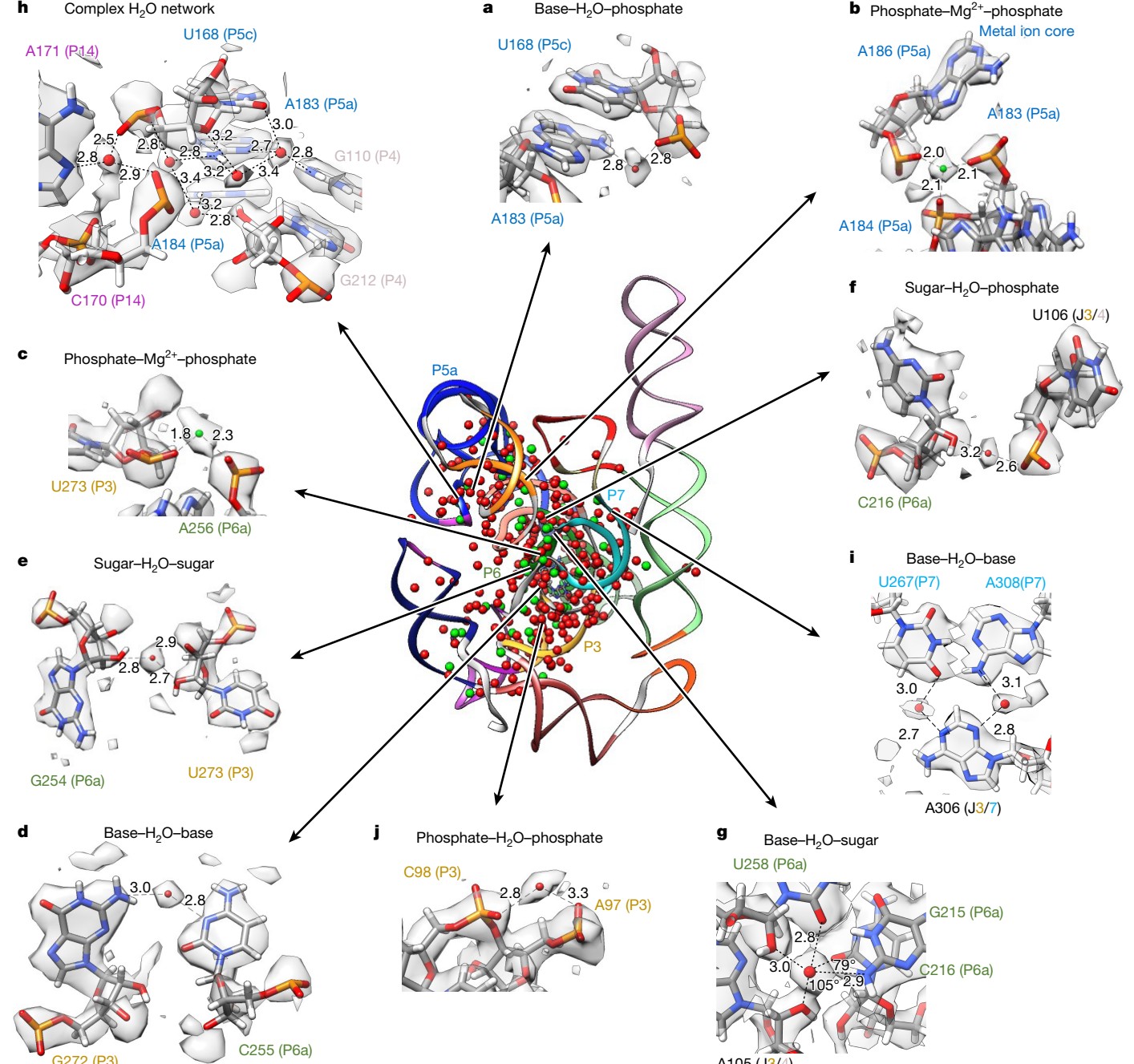

**Fig. 3 | Water and Mg²⁺ ion binding to nucleotides within and between domains.** Ribbon display of the model built in the 2.2 Å map (centre), colour-coded by domain as in Fig. 1, along with water (red spheres) and Mg²⁺ ions (green spheres). **a–j**, The panels on the outside highlight a selection of water and Mg²⁺ ions interactions, with nucleotide labels colour-coded by domain as in Fig. 1. All water and Mg²⁺ ions displayed are found in the 2.2 Å and 2.3 Å models; the same regions but with the 2.3 Å map and models can be found in Extended Data Fig. 6a–j. Distances (Å) from water and Mg²⁺ ions to RNA heavy atoms are labelled; only some of the contacts for each water and Mg²⁺ ion may be shown in each case. See Supplementary Video 1 to visualize panels **c**,**e**,**f**,**h** in 3D.

atoms, and distances to the nearest atom ranged between 2.5 Å and 3.5 Å, as per the SWIM criteria, with the most prevalent water–oxygen distance of approximately 2.8 Å (Extended Data Fig. 4n,o).

SWIM found many waters and Mg²⁺ ions that overlapped with those modelled in the lower-resolution cryo-EM structures of the full ribozyme[29,33–35] and X-ray structures of subdomains of the ribozyme[8,28,32] (Extended Data Fig. 5 and Supplementary Data, file 1). These previously modelled water and Mg²⁺ ions include those in the metal ion core of the well-characterized P4–P6 domain[8,32] (Fig. 3a–c and Extended Data Fig. 6a–c). Generally, the modelled Mg²⁺ ions bind to similar parts of nucleotides as Mg²⁺ ions modelled in previous RNA

studies[38] (Extended Data Fig. 7a,b,d), except for more frequent binding of Mg²⁺ ions to sugars in our SWIM model (see example in Extended Data Fig. 6k,l).

SWIM modelled many novel waters in our new structures. To investigate the confidence in their positioning and assignment, we took advantage of the availability of two discrete, independent, high-resolution cryo-EM maps. We examined whether the water modelled in both maps share equivalent locations; shared waters are referred to below as 'consensus' waters (Methods). Of the SWIM waters, 134 were identified as consensus, 53% and 48% of all SWIM waters in the 2.2 Å and 2.3 Å models, respectively (Fig. 2a,b).

Although all waters modelled were well resolved ($Q > 0.7$), consensus waters had a statistically significant higher $Q$ score than non-consensus waters (Fig. 2c) and were in regions of the RNA that were statistically significantly better resolved (higher $Q$ score; Fig. 2d) and more similar (smaller RMSD; Fig. 2e). Hence, we first analysed the binding motif of just the consensus waters, found in both the 2.2 Å and the 2.3 Å models, and then analysed the nature of the other, non-consensus, positions.

Consensus waters were bound throughout all parts of nucleotides in our cryo-EM structures (Extended Data Fig. 7a,c–e). However, some regions were more or less densely hydrated. The catalytic active site of *Tetrahymena* ribozyme had a similar number of waters per nucleotide as the other regions of the ribozyme (Supplementary Table 1), indicating that ordered waters may be important for tertiary interactions generally and there is not a particularly elaborate water structure in the active site when substrate groups are absent, as in our sample.

Indeed, all resolved tertiary interactions in our structures, which link distant nucleotides in the sequence into stable junctions, were extensively bound to well-positioned waters. For example, the catalytic active site is embedded in a highly conserved core with numerous non-canonical interactions. Stabilizing this core, where the P3 and P6a domains come in close proximity, the Watson–Crick–Franklin edge of C255 and the sugar edge of G272 meet in an orientation that is not favourable for direct hydrogen bonding. We observed an ordered water that bridges these two bases stabilizing the tertiary contact (Fig. 3d and Extended Data Fig. 6d). This water was not previously observed, although some signal was visible at low contours in the 3.1 Å cryo-EM map (Extended Data Fig. 6n). Another previously unidentified water stabilized these domains by bridging the sugars of G254 and U273 (Fig. 3e and Extended Data Fig. 6e).

Elsewhere in the conserved catalytic core, the backbone of P6a comes in close proximity to the junction that connects P3 and P4. We observed a chain of waters mediating this interaction. As an example, a water, not previously modelled, bridges the sugar of C216 and the phosphate of U106 (Fig. 3f and Extended Data Fig. 6f). A newly identified water binds to the G215–U258 wobble base pair of P6a in the minor groove. It forms hydrogen bonds with the O2 and O2′ of the U258 and the N2 of the G215 (Fig. 3g and Extended Data Fig. 6g). A water with the same binding atoms was previously highlighted in an unrelated RNA system within an A-form helix where it was observed to make an additional contact with the O2 of the downstream cytidine[39]. Our structure also contains a downstream cytidine, C216, with a long bond length of 3.3 Å from the O2 to the modelled water. However, C216-O2 makes an angle of 79° with the water and the G215-N6, too small for the tetrahedral arrangement of water (109.5°)[39]. A hydrogen bond is more likely formed with the O2′ of A105 in the junction between P3 and P4. Hence, this water may have a role in stabilizing the tertiary interaction between P6a, P3 and P4. This example illustrates how, to stabilize tertiary interactions, water networks can be structured differently from what is expected of a classic A-form helix.

Water has an integral role in stabilizing the tertiary structure generally. For example, in the region where P4, P5c and P14 come together, we observed at least five waters that form a network stabilizing this complex junction (Fig. 3h and Extended Data Fig. 6h). In the junction between P3 and P7, A306 is adjacent and in the same plane as the A308–U267 base pair, but instead of A306 forming direct RNA–RNA hydrogen bonds with the A–U base pair, the interaction is mediated by two waters (Fig. 3i and Extended Data Fig. 6i).

SWIM modelled waters interact with multiple phosphates, potentially having the role of shielding the negative charge typically fulfilled by ions. In the highly solvated and well-resolved P3 domain, a density peak is visible bridging phosphates of two consecutive nucleotides with distances of 2.8 Å and 3.3 Å (Fig. 3j and Extended Data Fig. 6j). No previous structures modelled a water or an ion at this site. These distances are too long for a $Mg^{2+}$ ion, which is typically approximately 2 Å away from a phosphate, and hence the atom is more likely a water.

However, we could not rule out a monovalent cation at this position (further discussed in Extended Data Fig. 6j–m). To get a better sense of the limitations of our modelling at 2.2 Å and 2.3 Å resolution, we analysed the SWIM assignments that disagreed between our two independent maps.

## Exploring non-consensus waters

We were left with roughly half of the automatically modelled waters that were identified by SWIM in only one of the two independently reconstructed cryo-EM maps. A similar percent overlap was observed in the two asymmetric units of the X-ray crystal structure of the P4–P6 subdomain[8] (Extended Data Fig. 5g). We examined the non-consensus waters in more detail by placing them into the map that they were not modelled in and observing the model and map features. If the disagreement is explained by noise or difference in solvent structure, we expected the models and map features to differ significantly between the two independent maps. For 30% of the non-consensus waters, a peak was modelled nearby but in a slightly different position in the other map, sometimes leading to disagreement in the assignment (for example, water in one model, a $Mg^{2+}$ ion in the other; Extended Data Fig. 8a–c). Of all waters, when placed in the other map, 84% still exhibited high cryo-EM density (more than 5$\sigma$) and resolvability ($Q > 0.7$; compared with 1.2% of randomly sampled positions in the solvent shell), and 75% of all waters additionally passed the half-map resolvability criteria (compared with 0.6% of randomly sampled positions) (Extended Data Figs. 4a–h and 8a,b). This suggested that many of the non-consensus waters had significant density features in both maps and generally that very few of the non-consensus waters originated from experimental noise or an actual difference in solvent structure. The above data led us to conclude that SWIM is reliable in locating density peaks and these peaks are unlikely to be noise.

Generally, non-consensus waters have a significantly lower peak density than consensus water (Fig. 4a), providing a simple explanation for the inconsistency of SWIM in modelling. To understand the origin of the lower peak densities, we reasoned that peak density can be reduced by two factors: occupancy (that is, how often water is present anywhere in the site) and high positional spread (that is, water diffuses within the binding sites and is not always localized to the peak coordinate; Fig. 4b). Cryo-EM maps at this resolution do not contain the information to neatly decompose these factors. To develop hypotheses for what accounts for lower peak densities of non-consensus SWIM-modelled waters, we carried out explicit solvent all-atom molecular dynamics (Extended Data Fig. 9 and Methods). The root mean square fluctuation (RMSF) of the RNA in 30 400-ns simulations was used to estimate the global flexibility of the ribozyme. A significant negative correlation exists between the $Q$ score and RMSF (Extended Data Fig. 3e–h), in agreement with the premise that inherent flexibility is the primary attribute explaining poor resolvability in peripheral regions of the ribozyme. These results indicate that cryo-EM freezing artefacts are not the major determinant of poor resolvability and simultaneously offer support for the credibility of the simulations.

Waters (simulated with the TIP4P-D water model[40]) were initially placed randomly and were not guided by the cryo-EM map during simulation to enable unbiased comparison between molecular dynamics and cryo-EM. To assess the reliability of making observations of water dynamics in the molecular dynamics simulations, we compared water-binding sites found in the molecular dynamics simulation to the SWIM-modelled cryo-EM waters. On average, the cryo-EM waters had a predicted molecular dynamics water concentration of 90 M (0.054 waters per Å³), and only 10% of cryo-EM waters had predicted molecular dynamics water density lower than bulk water (55 M, 0.033 waters per Å³; Fig. 4c and Extended Data Fig. 9e). As expected, molecular dynamics water-binding sites that were modelled by SWIM in both cryo-EM maps had higher water density in the molecular dynamics simulations

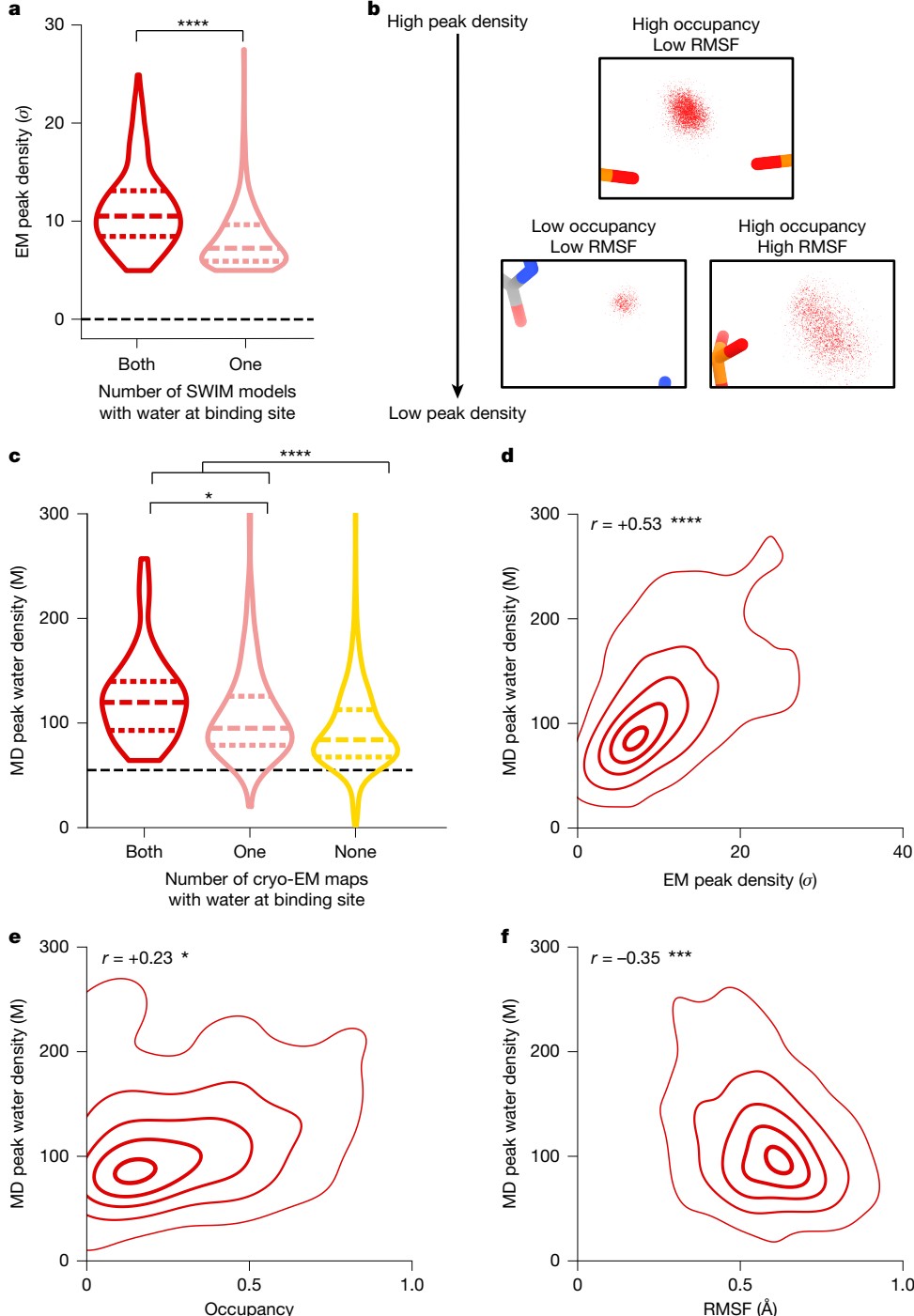

**Fig. 4 | Water dynamics in molecular dynamics simulation with reference to cryo-EM water-binding sites. a**, Cryo-EM peak density of all SWIM-identified waters is plotted for waters in both cryo-EM maps ('consensus' (red) $n = 268$) or only one cryo-EM map ('non-consensus' (pink) $n = 268$). The violin plot displays the range, dotted lines are the quartiles, and pairwise significance was determined by two-sided Mann-Whitney $U$-test ($P = 3.6 \times 10^{-26}$). The black dashed line is the mean density of the map. **b**, Example of molecular dynamics water-binding sites displaying how low occupancy or high RMSF can reduce the peak density. For every molecular dynamics frame, the position of water, if present, is a small red dot. **c,d**, Water-binding sites from molecular dynamics (MD) simulations are grouped by whether they are found in both cryo-EM maps (red; $n = 76$), only one cryo-EM map (pink; $n = 167$) or no cryo-EM maps (yellow; $n = 1,984$). The molecular dynamics and EM (2.2 Å map) peak water density are the maximum density within 1 Å of the molecular dynamics peak coordinate.

The molecular dynamics peak water density is displayed (**c**; one versus two cryo-EM maps: $P = 1.0 \times 10^{-4}$; one or two versus no cryo-EM maps: $P = 1.2 \times 10^{-14}$); statistics are as in panel **a**. The black dashed line is the density of bulk water. For each molecular dynamics water-binding site found in both or one cryo-EM map, the cryo-EM peak density is plotted against the molecular dynamics peak water density; the contours display the 5%, 25%, 50%, 75% and 95% probability densities from kernel density estimation (**d**). Pearson's correlation coefficient and $P$ value, two-sided hypothesis test for 0 correlation, are reported ($P = 3.2 \times 10^{-19}$). **e,f**, For each molecular dynamics water-binding site, the proportion of time that any water occupied that site (**e**; $P = 2.9 \times 10^{-4}$) and the RMSF of the waters found at that binding site (**f**; $P = 2.5 \times 10^{-8}$) across all simulations are plotted against the molecular dynamics peak water density, as displayed in panel **d**. Significance is reported as: *$P < 0.05$, ***$P < 10^{-6}$ and ****$P < 10^{-8}$.

than waters modelled in only one cryo-EM map or locations where SWIM did not model a water (Fig. 4d and Supplementary Data, file 2). In addition, the molecular dynamics peak density correlated with the cryo-EM peak density at molecular dynamics water-binding sites that were modelled by SWIM (Fig. 4c). Our observations thus increased confidence in the use of the simulation in interpreting cryo-EM data and suggesting hypotheses about the nature of water dynamics.

The molecular dynamics simulation enabled the examination of the two factors that we proposed could reduce peak density: occupancy and positional spread. The variation in the molecular dynamics peak density is partially accounted for by the variation in occupancy and positional spread (RMSF) of water (Fig. 4e,f), suggesting that both factors contribute to reduction in peak density, which results in the deviations in the SWIM-based water modelling for the two maps. The inspection of sites with higher positional spread led us to analyse water maps in a method independent of discrete peak identification by SWIM, described next.

## Complex and diffuse water networks

On the basis of the results above, we expanded our analysis beyond highly ordered waters traditionally modelled in peak densities by SWIM and other algorithms[41]. We visually observed patches of density that diffuse into specific directions and are correlated between the two independent maps. We next explored whether the cryo-EM density could contain information about more diffuse and unmodelled water networks.

We first considered whether the diffuseness of density might be due to a lack of resolution. Comparison with a lower-resolution (3.1 Å) cryo-EM map showed that in many instances, although the density was too diffuse to model at 3.1 Å, the signal for waters now resolved at 2.2 Å was present (Extended Data Fig. 6n–p). We expected similar effects at a smaller distance scale for the 2.2 Å map, for example, blurring the density between the $Mg^{2+}$ ion and coordinating waters (Extended Data Fig. 8c). The resolution-dependent diffuseness of cryo-EM density suggests caution on automatically assigning peaks to water or $Mg^{2+}$ ions at 2.2 Å without cross-validation, as was done here, or manual inspection. However, the confirmation that diffuse density in the 3.1 Å map overlaps with modelled water, confirmed in our 2.2 Å and 2.3 Å maps, motivated further investigation of these diffuse densities.

Globally, we observed excellent agreement between the 2.2 Å and 2.3 Å maps in the cryo-EM density of the solvent shell (Fig. 5a; displayed $3\sigma$ above background). We observed wires of diffuse density that are present in the 2.2 Å and 2.3 Å maps, as well as the predicted molecular dynamics water density (Fig. 5a). Taking a slice of density, we observed water networks along the grooves of helices as well as around the backbone, particularly in the region of tertiary contacts (Fig. 5b). Zooming in further, around the backbone of P6a, we observed a complex network of waters (Fig. 5c). The placement of waters from SWIM matched well between the 2.2 Å and 2.3 Å maps: eight waters were modelled in similar positions. Six of the overlapping water positions further overlapped with a local maximum in the simulated molecular dynamics water density. This suggested that molecular dynamics is capable of predicting some locally ordered waters. The SWIM-assigned water positions disagreed in five positions between the two independent maps; however, at all of these positions, the cryo-EM maps in the solvent shell were highly similar. Moreover, the cryo-EM and simulated molecular dynamics densities agreed at low contours, suggesting that the cryo-EM density maps and molecular dynamics simulation agreed not only on many of the ordered water-binding sites but also on the diffuseness of water through a network. This was even the case for regions where map diffuseness precluded atomic water placement by SWIM (Fig. 5c).

Examining near the catalytic core, specifically, the cavity created by the major groove of P7, A261, C261 and A306, we saw diffuse densities that agree well between the two cryo-EM maps, including the pocket that would bind to the guanosine substrate ωG of the ribozyme (Fig. 5d). Turning towards the substrate-binding site, we again observed similar density features in the cryo-EM maps; diffuse densities, with the same structured network between the two maps (Fig. 5e). In both sites, molecular dynamics predicted less hydration density than at other sites; although the RNA is fully solvated at each point in the simulation, molecular dynamics predicted that this solvation shell is more dynamic. This cavity is directly exposed to bulk solvent, a potential explanation for why the molecular dynamics predicted these sites to have less-ordered waters than observed experimentally in flash-frozen samples. Analysing a nearby, less-solvent exposed, tertiary interaction between the backbone of P9.1 and P7, molecular dynamics predicted a water wire between these backbones, which is also supported in the cryo-EM densities (Fig. 5f). The SWIM models disagreed on some water placements in this region due to the diffuseness of the density. This region may be more accurately modelled by an ensemble of partially occupied water positions as opposed to a few specific positions.

These examples reveal that the cryo-EM maps contained information about the solvent structure that is not currently modelled in the traditional atomic structure representation underlying SWIM and depositions in the Protein Data Bank (PDB). We quantified the agreement by comparing the distribution of density in the solvent shell from SWIM models and molecular dynamics with the 2.2 Å map. Despite conservative modelling, the SWIM model obtained a better cross-correlation coefficient than molecular dynamics, although both are far from our level of experimental precision, here estimated by comparison between the 2.2 Å and 2.3 Å maps (Supplementary Table 2).

To delve into this comparison further, we treated the density as a classification task with the goal of identifying all the voxels with a significant density (more than $3\sigma$) in the well-resolved (RNA $Q > 0.6$) solvent shell (1.8–3.5 Å away from RNA) of the 2.2 Å map. The 2.3 Å map at $3\sigma$ was able to match 71% of these solvent locations, also known as recall, with a low false positive rate of 5.6% and a high precision of 63% (Extended Data Fig. 8d–f). At the same recall (71%), the molecular dynamics water density falsely identified 35% of the voxels and had a much lower precision of 22%, revealing a large accuracy gap for molecular dynamics (Extended Data Fig. 8d–f). Furthermore, this analysis confirms that the SWIM-identified waters and $Mg^{2+}$ ions only accounted for a minor proportion of the consensus density in the solvent shell. Comparing the SWIM-identified waters to the cryo-EM density, we observed high precision; SWIM-identified waters recovered 10% of the solvent positions with a precision of 96%, whereas molecular dynamics recovered this many with a precision of only 68%. However, there was a large proportion of density left unmodelled by SWIM, plateauing at approximately 25% solvent locations recovered (Extended Data Fig. 8d–f). This low recall was due to the SWIM criteria being designed to only detect rigidly ordered waters and ions, and not other solvent and ion densities. Overall, using classification performance summary metrics, we observed good experimental reproducibility, with reasonable performance for the SWIM model and poorer performance for molecular dynamics predictions. Finally, we used this metric to assess the local accuracy of the TIP4P-D model in predicting the water structure by direct comparison to the cryo-EM map (Extended Data Fig. 8g–i). The molecular dynamics-predicted water structure agrees with the cryo-EM map most in the peripheral tetraloop–tetraloop receptor, but performs poorly in other regions, most notably in the catalytic core (mean normalized nucleotide area under the precision-recall curve of 0.58 and 0.34, respectively; Supplementary Table 1).

## Discussion

This investigation was made possible by the high quality of the cryo-EM map that we obtained, which, to our knowledge, is the highest resolution reported for an RNA-only system to date. In the future, with the development of newer methods in cryo-specimen preparation[42,43],

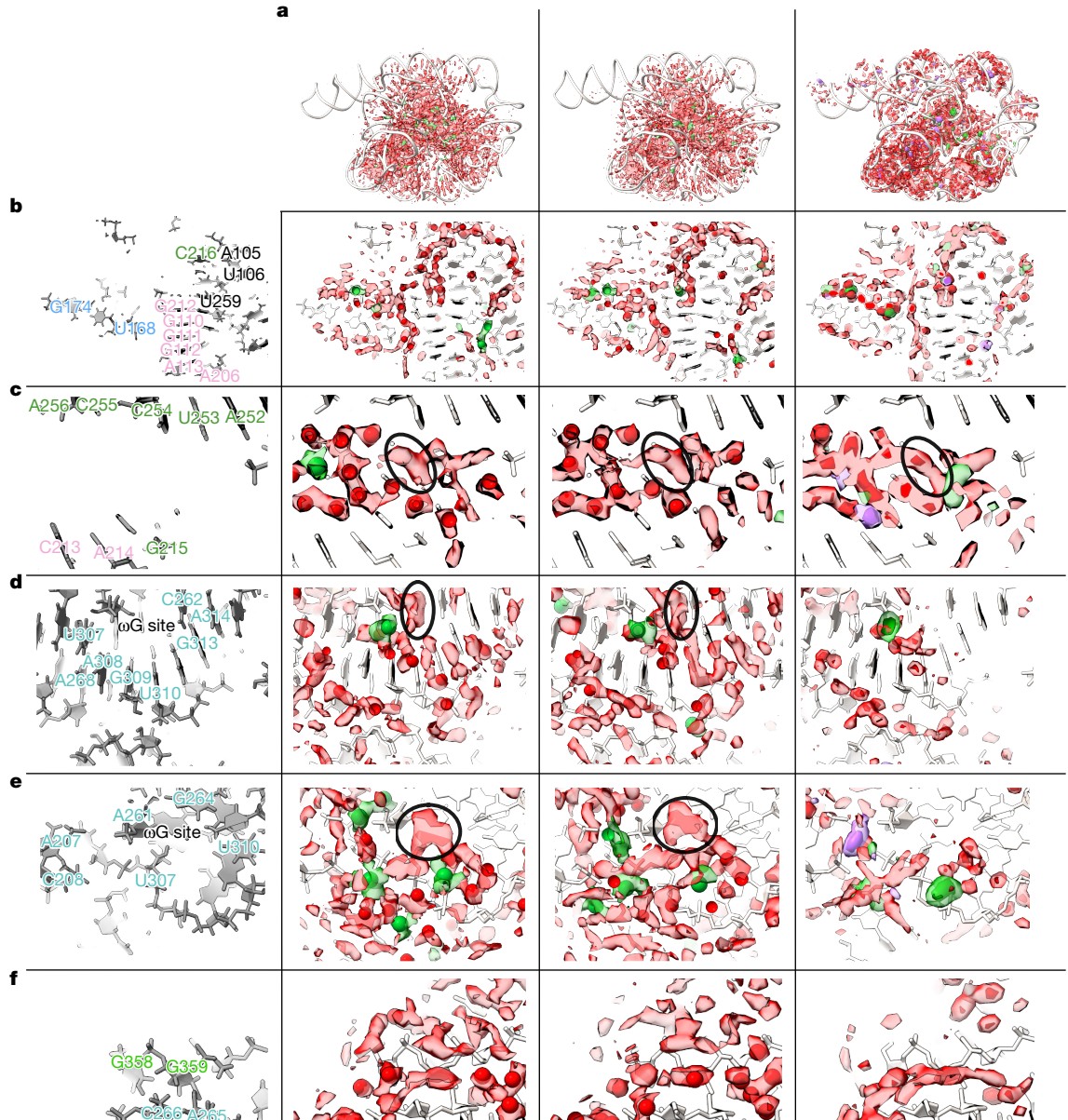

**Fig. 5 | Diffuse cryo-EM densities and molecular dynamics water networks.**
Comparison of density features surrounding the RNA (red) between the cryo-EM maps and molecular dynamics. The first column labels the nucleotides in the region, coloured according to domains from Fig. 1. The second and third columns are the 2.2 Å and 2.3 Å cryo-EM densities, respectively, at $3\sigma$ at least 1.8 Å away from the RNA coloured in transparent green (within 2.5 Å of a $Mg^{2+}$ ion) and red (remainder of density). The models are displayed in white, with the SWIM modelled $Mg^{2+}$ ion as green spheres and water as red spheres. The final column is the predicted density from molecular dynamics after local alignment (Methods). Water (dark red at 101 M, light red at 68 M), $Mg^{2+}$ ion (dark green at 46 M, light green at 14 M) and $Na^+$ ion (dark purple at 28 M, light purple at 14 M)

densities are displayed. The 2.2 Å RNA is pictured in white for reference.
**a**, Densities of water and ions around the RNA for a global view of the ribozyme. **b**–**f**, Specific regions of interest, showing water networks along the groove of helices and around the backbone (**b**), a network of waters around the backbone of P6a (**c**), water networks surrounding the catalytic core (**d**,**e**) and a water wire bridging the backbone of two RNA helices (**f**). The black circle (**c**) highlights a density feature in cryo-EM and molecular dynamics simulations whose irregular shape precluded SWIM assignment of an atomically ordered water. A density of unknown identity in the binding site for the guanine substrate is circled in black (**d**,**e**). See Supplementary Video 2 to visualize the 3D context of panels **b**–**e**.

larger datasets and improved data-processing algorithms that account for local flexibility[44,45], RNA cryo-EM maps have the potential to be extended closer to atomic resolution. With high-quality density maps and an optimized automated SWIM algorithm, water positions were automatically modelled based on the strength of the local signal in the map, concurrent resolvability in both full and half maps, and physically plausible distances from RNA atoms. The SWIM-modelled waters were included in the PDB-deposited model. The SWIM criteria were shown to be stringent, with only approximately 1% of randomly sampled positions in the solvent shell meeting the criteria. However, at approximately 2 Å

resolution, there remained ambiguity in precise peak placement and molecular identification to be either water or cations in some instances. Therefore, a future challenge is to develop improved algorithms to reliably characterize the identity of bound water and ions in those instances at this resolution range. Beyond the geometry of coordination, such an algorithm may take biophysical properties of electron scattering of cations into account to differentiate between waters and ions[46], especially monovalent ions. Fortunately, in our study, cross-validation was facilitated by two independent high-resolution maps of *Tetrahymena* ribozyme, which were highly similar in the well-resolved regions.

Contrary to the common portrayal of water and ions as molecules dispersed on the periphery of biomolecules, in the full ribozyme in vitreous ice, we observed water and $Mg^{2+}$ ions principally in the interior, as also observed in the crystal structure of the P4–P6 domain[8]. We not only validated our finding for well-studied water and ions, such as those previously known in the metal ion core[32], but also uncovered many other novel waters that formed bridges between RNA nucleotides stabilizing the compact fold of ribozymes (Fig. 3). We resolved a structured water network surrounding the active site of the ribozyme, despite the site being solvent exposed (Fig. 5d,e). These ordered waters could be important for the pre-structuring of catalytically important ions or to reduce the entropic penalty of desolvation upon substrate binding and may be important for understanding currently unexplained effects from nucleotide analogue interference mapping experiments[47–52] (Supplementary Table 3). To fully appreciate the biophysical information that can be derived from the cryo-ensemble of water structure, future study of the effect of radiation damage and the freezing process on water bound to RNA is warranted[53,54].

Owing to our conservative criteria in the models, there remained signals in the map that are not annotated in the PDB-deposited model. For example, we only modelled enough ions to neutralize around one-quarter of the RNA charge, implying there are some aspects of ionic composition not represented in our structural models. This missing representation of ions is pervasive in all experimental structures, which only model ordered waters and ions. Furthermore, our consensus criteria revealed that only half of the automatically modelled waters agree and have been modelled, despite high agreement between cryo-EM densities (Fig. 2a,b).

We turned to molecular dynamics simulation to develop hypotheses for the biophysical explanation of the unmodelled water cryo-EM densities. Comparisons with molecular dynamics simulations supported the hypothesis that the cryo-EM map SWIM-modelled peaks capture ordered waters that have high occupancy and low positional spread (Fig. 4). Furthermore, some diffuse cryo-EM densities matched diffuse water networks in the molecular dynamics simulations, supporting the hypothesis that cryo-EM may resolve diffuse density representing flexible water networks (Fig. 5). If this is true, even if RNA can be resolved at higher resolution, some or most water density will remain diffuse. In the future, we foresee two methods by which these networks may be understood in more detail. First, new methods of heterogeneity analysis of individual cryo-EM particles that contain high-resolution information should be developed and validated, enabling the visualization of water networks via the ensemble of water placements revealed. Second, although the simulation of water dynamics cannot yet accurately model experimental RNA and solvent structural fluctuations fully, the observed agreement warrants further investigation into the utility of molecular dynamics in interpreting and predicting water structure around RNA, including various catalytic states of the *Tetrahymena* ribozyme. In the future, cryo-EM data could be used to evaluate and compare simulation methods, including quantum mechanical simulations and alternative force fields, with particular focus on different parameterizations of water and ions[55–58].

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

## Methods

### Cryo-EM sample preparation and data collection

*Tetrahymena* ribozyme RNA (125 kDa, the linear L-21 ScaI ribozyme spanning residues 22–409 without the P1 substrate or ωG) was prepared as previously described[29]. The ribozyme sample (approximately 25 μM, approximately 3 mg ml⁻¹) in 50 mM Na-HEPES pH 8.0 was denatured at 90 °C for 3 min and cooled to room temperature for 10 min. MgCl₂ was then added to a final concentration of 10 mM, and the samples were incubated at 50 °C for 30 min. The samples were cooled again to room temperature for 10 min. Three microlitre volumes were applied onto glow-discharged 200-mesh R2/1 Quantifoil copper grids. The grids were blotted for 4 s and rapidly cryocooled in liquid ethane using a Vitrobot Mark IV (Thermo Fisher Scientific) at 4 °C and approximately 100% humidity. The grids were screened using a Talos Arctica cryo-electron microscope (Thermo Fisher Scientific) operated at 200 kV. The grids were imaged in a Titan Krios G3i cryo-electron microscope (Thermo Fisher Scientific) operated at 300 kV at a magnification of ×105,000 (corresponding to a calibrated sampling of 0.82 Å per pixel). Micrographs were recorded by EPU software (Thermo Fisher Scientific, v2.7) using 'Faster Acquisition' with a throughput of approximately 420 movie stacks per hour with a Gatan K3 Summit direct electron detector, where each image was composed of 30 individual frames with an exposure time of 2.5 s and a dose rate of 22.9 e⁻ s⁻¹ Å⁻². Finally, a total of 18,365 movie stacks was collected with a defocus range of −0.5 to −2.0 μm.

### Image processing

All micrographs were motion-corrected using MotionCor2 (ref. 60), and the contrast transfer function (CTF) was determined using CTFFIND4 (ref. 61). All particles were autopicked using the NeuralNet option in EMAN2 (ref. 62) and further checked manually. The resulting number of boxed particles was 3,804,753. Then, particle coordinates were imported to Relion[63], where three rounds of 2D classification were performed to remove 2D class averages with less resolved features. The selected 1,823,256 particles were imported to cryoSPARC[64] for generating ab initio maps, and a good map with clear RNA features was derived. Then, starting with this map, non-uniform refinement[65] together with local and global CTF refinement was performed, yielding a map with 2.2 Å resolution from 1,181,331 particles. Further 3D variability analysis[66] was performed to classify slightly different conformations, and two conformations were obtained. Final maps were achieved after another round of non-uniform refinement[65] for each of the two classes, 708,006 and 473,325 particles, respectively, with resolutions at 2.2 Å and 2.3 Å, respectively. The cited resolutions for the final maps were estimated by the 0.143 criterion of the FSC curve in cryoSPARC. The local resolution map was calculated using cryoSPARC local resolution estimation using default values. See more information in Extended Data Table 1 and Extended Data Fig. 1.

### Map sharpening

The reconstructed maps for both 2.2 Å and 2.3 Å were further processed with phenix.auto_sharpen[67]. Half-maps were also sharpened for modelling waters and ions. In phenix.auto_sharpen, the amount of sharpening was reflected by the b_sharpen parameter. For the 2.2 Å map, the b_sharpen applied was 79.23, giving a final b_iso of 20.00 (initial 99.23), and for the 2.3 Å map, b_sharpen was 74.07, giving a b_iso of 30.45 (initial 94.07). The b_iso indicates how quickly amplitudes fall off with increasing frequencies in Fourier space. In theory, the higher b_iso of the 2.3 Å map could be attributed to (1) lower signal-to-noise ratio (SNR) due to fewer particles, and/or (2) more varied particle conformations with larger atomic displacements than in the 2.2 Å map.

### Model building

The atomic model of full-length apo *Tetrahymena* ribozyme (PDB ID: 7ez0)[29] was first rigidly fitted into 2.2 Å and 2.3 Å sharpened maps, respectively. The resultant models were refined into the sharpened maps using phenix.real_space_refine with secondary structure and geometry restraints, using the default parameters and five cycles[68]. The models were then further refined into the sharpened maps with ISOLDE in ChimeraX (v1.6.1)[68–70], which improved model geometry and clashscore. SWIM was then applied to model ions and water molecules (see below). The quality of the final models was evaluated by MolProbity[71] and Q score[31]. Statistics of the map reconstruction and model optimization are summarized in Extended Data Table 1. All figures were made using Chimera[72] and ChimeraX[70].

### SWIM

The SWIM procedure, which is available on GitHub (https://github.com/gregdp/segger (v2.9.7); and scripts (https://github.com/DasLab/Water-CryoEM-ribozyme)), identified peaks in density and modelled these peaks as water, Mg²⁺ ion or no atom. Previously, SWIM was run on a 1.34 Å map of apoferritin, using a 2σ density threshold and with no Q score or half-map threshold[17]. In this study, the density threshold had been increased to 5σ and minimum Q score criteria were imposed, including Q scores in full and half-maps and Q scores of bound nucleotides in the full maps for increased stringency. Specifically, SWIM was applied to the 2.2 Å and 2.3 Å cryo-EM sharpened maps, using sharpened half-maps within the procedure, and the refined models with an additional guanine placed in the guanine-binding site, which was subsequently removed, as follows:

(1) The average (avgD) and standard deviation (σ) of all density values in the full map were calculated.

(2) The full map was segmented using Segger, with no grouping steps, at a threshold of 3σ above avgD. Each segment was then considered in order of decreasing volume.

(3) For each segment, the point within the segment that had the highest interpolated density value ($P_{max}$) was identified, using the Fitmap.locate_maximum function in Chimera.

(4) The following rules were applied to determine whether to ignore the segment, or model a water molecule or an ion at $P_{max}$:

(i) The Q score at $P_{max}$ was calculated in the full map and the two half-maps; half-maps have previously been used to model water[73]. Only segments with a Q score at $P_{max}$ above the chosen value Q_peak_min of 0.7 in all three maps were further considered. Choosing a higher Q_peak_min reflects a better-resolved water or ion.

(ii) If the atom nearest to $P_{max}$ was in a poorly resolved nucleotide (Q < 0.6), no water or ion was modelled. We refer to this parameter as Q_res_min.

(iii) The density value at $P_{max}$ was calculated by tri-linear interpolation. If the density value was less than 5σ above avgD, no water or ion was modelled.

(iv) If there were non-polar atoms (carbon atoms or phosphorus) that would clash with $P_{max}$ (within 3.2 Å), no water was modelled. Likewise, for ions, if there was a carbon atom that would clash with $P_{max}$ (within 3.0 Å), no ion was placed.

(v) If, as expected for a cation, electronegative atoms neighboured $P_{max}$ but no electropositive atoms that would repel a cation neighboured $P_{max}$, a Mg²⁺ ion was modelled. Electronegative atoms were all oxygen atoms and non-protonated nitrogen atoms within 1.8–2.5 Å of $P_{max}$. Electropositive atoms were protonated nitrogen atoms within 1.8–3.4 Å of $P_{max}$.

(vi) Otherwise, if there were RNA atoms that could hydrogen bond with $P_{max}$ and/or ions close to $P_{max}$, a water molecule was modelled. Specifically, any nitrogen or oxygen atoms 2.5–3.4 Å away from $P_{max}$ or an Mg²⁺ ion 1.8–2.5 Å away from $P_{max}$.

(vii) When a new water was modelled, if it was within minWaterD (2.5 Å) of an already modelled water molecule, it was ignored because it was too close. This water may be an 'alternate conformer' of the same water, which could be modelled in partial occupancy, but that was not done for this study. This was also done for Mg²⁺ ions, using minIonD (4.5 Å).

(5) The method was repeated until no more water or $Mg^{2+}$ ions were modelled; in total three iterations were performed.

Only $Mg^{2+}$ ions bound directly to the RNA and water were modelled. It is conceivable that our models may have missed or mis-assigned monovalent ions, such as $Na^+$ and $Cl^-$, as well as $Mg^{2+}$ ions interacting with RNA through water (see discussion in Extended Data Figs. 6k–m and 9e,f).

### Consensus analysis

We referred to high-confidence waters if they are found in both the 2.2 Å and the 2.3 Å models; there is consensus. Consensus was defined by passing two criteria concurrently: superimposable and the same binding site, ensuring we captured waters and $Mg^{2+}$ ions that are spatially superimposable and have the same local chemical environment. Superimposable waters were within 1 Å of one another after the alignment of local RNA atoms. Local RNA atoms were all heavy atoms in nucleotides that contained an atom that was 10 Å or closer to either water. The same binding site was defined as follows. For each water in the pair of water being compared, the set of 'close RNA atoms', defined as RNA atoms 2.5–3.2 Å away from the water, was obtained, expanding by 0.3 Å if no RNA atoms were within 3.2 Å. If the same close RNA atoms were found within an expanded distance 2.5–3.5 Å of the other water and vice versa, then the pair of waters were said to have the same binding site. The distance was expanded to account for distance uncertainties. If there were a pair of waters, for example, one in the 2.2 Å model and another in the 2.3 Å model, that passed both criteria, the waters were considered consensus waters, otherwise, they were non-consensus waters. The same criteria were used for $Mg^{2+}$ ions with a close binder distance of 1.8–2.2 Å and an expanded distance of 1.8–2.5 Å. Non-consensus waters and $Mg^{2+}$ ions were modelled by SWIM but do not meet the very strict requirement of being superimposable and binding to the same RNA sites in the 2.2 Å and 2.3 Å maps.

Separate models were deposited in the PDB with consensus water and ions (9CBU and 9CBW for the 2.2 Å and 2.3 Å models, respectively) and a different model with all automated water and ions (9CBX and 9CBY for the 2.2 Å and 2.3 Å models, respectively). For confident water and ion placement only, the first model should be used. The B-factors reported are calculated according to the formula $B = 150(1 - Q)$ (ref. 18).

These consensus criteria were used identically to assess overlap with previous X-ray and cryo-EM models (PDB IDs: 7EZ0, 7EZ2, 7R6L, 7XD5, 7XD6, 7XD7, 7YG8, 7YG9, 7YGA, 7YGB, 7YGB, 8I7N, 1GID, 1HR2 and 1X8W). All previous structures were compared when manually inspected for sequence alignment. For comparison to multiple models (Extended Data Fig. 5), waters and $Mg^{2+}$ ions from all models were combined, including those that overlap. For the '2.2 Å and 2.3 Å' category, only the consensus waters and $Mg^{2+}$ ions were used.

### Analysis of models

For all analysis and averaging, hydrogens were ignored, only heavy atoms were considered. Map values were calculated using tri-linear interpolation. Map resolvability was quantified by a map-to-model measure of atomic resolvability, the $Q$ score, which is independent of the contour level[31]. All $Q$ score calculations, including those with half-maps, were calculated using sharpened maps and default parameters in the command line tool (https://github.com/gregdp/mapq). Per-residue $Q$ scores were calculated by averaging across all heavy atoms in each residue without mass-weighting as is the default in $Q$ score. The expected $Q$ score was calculated using the previously published linear relationship[31]. The RMSD between the two models was calculated after the superposition of all RNA-heavy atoms. Distances to RNA atoms were calculated with custom scripts.

### Sampling of positions in the solvent shell

For assessment of the strictness of the SWIM criteria, positions in the solvent shell of the RNA were sampled and each position was assessed by the SWIM criteria. For sampling the positions, a cube containing the RNA with a point every 1.67 Å, only keeping points that were between 1.5–3.5 Å from an RNA heavy atom in a well-resolved nucleotide ($Q > 0.6$).

For assessing how well SWIM-modelled waters fit in the alternative map, the map they were not modelled in, the waters were placed in the other map after locally aligning the RNA (RNA nucleotides within 10 Å). The SWIM criteria were assessed for these positions as above.

### Molecular dynamics

Details and input files are available on GitHub (https://github.com/DasLab/Water-CryoEM-ribozyme) and simulation files can be found in the Stanford Digital Repository (https://doi.org/10.25740/sw275qs6749). Atomic RNA coordinates were taken from the 2.2 Å model. To ensure RNA-folding stability during the simulation, 20 $Mg^{2+}$ ions bound to specific RNA sites were pre-positioned in the initial conditions of the simulation; these were modelled by an earlier version of SWIM in both the 2.2 and the 2.3 Å maps. During the simulation, some of these 20 ions moved, whereas some stayed in their placed positions throughout the simulation. As $Mg^{2+}$ ion placement by molecular dynamics is necessary but biased (that is, not randomly sampling $Mg^{2+}$ ion-binding sites), the comparison between molecular dynamics and cryo-EM $Mg^{2+}$ ion-binding sites was not analysed in detail (Supplementary Data, file 3). Seventy-five additional $Mg^{2+}$ and 196 $Na^+$ were added to neutralize the system. For six simulations of each force field, these additional ions were added randomly, whereas for four simulations, they were added using the Coulombic potential-guided placement method 'addIons' in LEaP[74]. Three force fields were used, DESRES[75], parmBSC0χOL3 using mMg and parmBSC0χOL3 using nMg[58,76–79]. mMg and nMg were parameterized using an in-house parameter file and placed in a TIP4P-D octahedral box. For each independent simulation, the system was minimized with 500 steps of steepest descent followed by 500 steps of conjugate gradient descent three times. Harmonic restraints of 25, 20 and 15 kcal mol$^{-1}$ Å$^{-2}$ were used on the RNA for first, second and third minimization, respectively. The system was then heated from 0 K to 100 K over 12.5 ps, harmonically restraining the RNA with a restraint of 15 kcal mol$^{-1}$ Å$^{-2}$. The system was further heated with the same restraints from 100 K to 310 K over 125 ps. All simulations were run on a single graphical processing unit (GPU) using the Amber20 Compute Unified Device Architecture (CUDA) version of particle-mesh Ewald molecular dynamics (PMEMD)[80]. The system was equilibrated with harmonic restraints on RNA atoms for 30 ns. The restraint strength started at 15 kcal mol$^{-1}$ Å$^{-2}$, was reduced by 2 kcal mol$^{-1}$ Å$^{-2}$ every 1 ns for the first 5 ns, then by 1 kcal mol$^{-1}$ Å$^{-2}$ every 1 ns for the next 5 ns and finally it was reduced by 0.1 kcal mol$^{-1}$ Å$^{-2}$ every 2 ns for the final 20 ns. Production simulations were performed (without restraints) at 310 K and 1 bar using the NPT ensemble, a Berendsen thermostat and a barostat. Every 200 ps, snapshots were saved. All simulations were run for 400 ns. These simulations used a 2-fs step, and bond lengths to hydrogen atoms were constrained using SHAKE. The cut-off for non-bonded interactions was at 9 Å.

### Molecular dynamics analysis

Scripts for all analyses are available on GitHub (https://github.com/DasLab/Water-CryoEM-ribozyme). The simulation was autoimaged, and 1-ns frames were aligned using all RNA heavy atoms using CPPTRAJ. Analysis was conducted on snapshots sampled every 1 ns; to reduce file size, water was cut-off at 3.5 Å. The simulations were further analysed using custom scripts with MDAnalysis[81,82] including RMSF and distance calculations. Throughout calculations, OP1 and OP2 were considered OP, and hydrogens were ignored.

For assessing the stability of the simulations generally, the RMSD from the 2.2 Å and 2.3 Å model was calculated from all RNA atoms after aligning to the ribozyme core defined as residues 31–42, 46–56, 96–102, 107–112, 116–121, 200–205, 208–214, 262–268, 272–278, 307-315-316,

318–331 and 405–406. For assessing the stability of the secondary structure, PDB-formatted coordinates of each frame were extracted using MDAnalysis, and information regarding base pairing was obtained from Rosetta rna_motif. Watson–Crick–Franklin base pairs from the model were analysed and the percentage frames in which they were maintained were reported using an in-house script. Ribodraw[83] was used to draw the secondary structure diagram of the starting coordinates. The interactions were manually coloured according to the percentage of frames in which the interaction was present.

RMSF and B-factor of RNA nucleotides were calculated per simulation by aligning to an average structure based on all RNA heavy atoms using MDAnalysis. The per nucleotide calculations were non-mass-weighted averages to match the $Q$ score, which does not weight the average $Q$ score by mass. Molecular dynamics-simulated maps were calculated using ChimeraX molmap with a resolution of 2.2 Å of all frames for each simulation independently. $Q$ scores were also calculated in these molecular dynamics-simulated maps and the average structures, as used in RMSF calculation, for each simulation to obtain 'simQ'.

Molecular dynamics binding sites were defined using the densities generated from all simulations (see below). Peaks in the molecular dynamics densities for water, $Na^+$ and $Mg^{2+}$ ions were identified with Segger. These were filtered to exclude peaks closer than 1 Å to another peak, choosing the peak with the highest density value. Every frame of the 30 simulations was analysed to identify waters and ions, whether they were in a binding site, and what their positions was in each binding site. First, for each nucleotide, a neighbourhood was defined as the set of nucleotides within 10 Å. Second, each frame of the simulation was locally aligned to each of these neighbourhoods, saving the water and ion positions in this locally aligned frame. The set of RNA atoms each water and ion was bound to in each frame was also saved. Third, for each frame, all neighbourhoods were combined. For waters present in more than one neighbourhood, the coordinates were selected from the neighbourhood where the water or ion was closest to the neighbourhood centre. Fourth, each water was labelled with a molecular dynamics binding peak if it was within 2 Å of the molecular dynamics peak coordinate and was bound to the same RNA atoms.

A molecular dynamics binding site was said to overlap with a cryo-EM SWIM-identified water or $Mg^{2+}$ ion if they bound the same RNA atoms and were within 2 Å of the molecular dynamics peak position.

Using the list of molecular dynamics binding sites, water occupancy for each simulation was defined as the fraction of frames in which water was bound in that site, as defined above, after removing frames in which the RNA deviated more than 3.4 Å from either cryo-EM structure. Water RMSF of each molecular dynamics binding site was calculated by taking the aligned coordinates of all waters in that binding site and calculating the RMSF of that set of coordinates.

The summary statistics for RNA atom binders (Extended Data Fig. 7) were calculated using MDAnalysis to calculate the distance between all RNA heavy atoms and the ions or waters. Distance cut-offs of 1.8–2.5 Å and 2.5–3.5 Å were used for ions and waters, respectively. Only the binding sites that had a water or $Mg^{2+}$ ion that was present for at least 10 frames (10 ns) were counted. The MgRNA values were counted (https://csgid.org/metalnas/), only counting inner-shell coordination.

For $Mg^{2+}$ ions, the residence time was defined as the time a single $Mg^{2+}$ ion was found at that binding site, allowing for 1 frame skip.

## Density comparison

To obtain the solvent density of the molecular dynamics simulation, first, local regions of interest were isolated by obtaining the list of RNA nucleotides that were within 10 Å of each modelled water or ion in the 2.2 Å and 2.3 Å map. For each set of nucleotides, the simulation frames were aligned on the heavy atoms of that set of nucleotides, and then the RNA, water, $Mg^{2+}$ ion and $Na^+$ ion density were calculated separately using DensityAnalysis from MDAnalysis. The densities were averaged across the 30 simulations for each local region. Then, to combine all regions into a composite map, at each voxel, the density value for all submaps whose centre was within 10 Å were averaged, weighted by the inverse of the distance from the centre of the submap to the voxel whose density was being calculated.

To obtain the average density around each nucleotide type (A, C, G or U), the transformation to align all ribozyme nucleobases to their respective reference nucleobase was calculated. Then, for each transform, the 2.2 Å resolution map (zoned > 1.8 Å from RNA heavy atoms) and the molecular dynamics water density (averaged as described above) were transformed, and transforms across each nucleotide type (A, C, G or U) were averaged evenly.

To assess overall agreement of the densities studied, the 2.2 Å density in the region 1.8–3.5 Å from all well-resolved RNA atoms ($Q > 0.6$) was used as reference. The comparison maps were (1) the 'independent map', the 2.3 Å map after transforming according to the alignment of the well-resolved RNA atoms ($Q > 0.6$) of the 2.2 Å and 2.3 Å models, (2) the 2.2 Å model molmap calculated in ChimeraX using 2.2 Å as the resolution; the water density was added to the $Mg^{2+}$ density weighted by 1.5, (3) the molecular dynamics density with water, $Mg^{2+}$ and $Na^+$ density summed with weight of 1.0, 1.5 and 1.3, respectively, densities are described above, and (4) the 'random map', the 2.2 Å map with all density values in the solvent shell randomly shuffled. The density was normalized by $Z$-score, and then the voxels in this region were isolated. The cross-correlation coefficient and mutual information were calculated as defined by Vasishtan and Topf[84], using 20 density bins for mutual information.

For the global comparison of solvent density, the modelling of water and ions was cast as a classification task. Therefore, the prediction task becomes predicting when the density is above a certain threshold. For this study, the 2.2 Å map was used as the reference map. The region 1.8.–3.5 Å from all well-resolved RNA atoms ($Q > 0.6$) was taken as the data, and all voxels above $3\sigma$ were defined as 'positives' and all other voxels as 'negatives'. The comparison maps were as above. The threshold of these comparison maps was varied to assess the classification accuracy. The precision (fraction of voxels above the current threshold for the comparison map that were correctly classified as positives), recall (fraction of all positives that were correctly classified above the current threshold for the comparison map) and false-positive rate (fraction of all negatives that were falsely classified as above the current threshold for the comparison map) at various thresholds were calculated. From these values, the precision-recall curve (PRC) and receiver operating characteristic (ROC) were plotted, and the area under the curve (AUC) was calculated. Owing to the imbalance of classes (most of the map is empty), the AU-PRC is the preferred measurement. Finally, the maximal Matthews correlation coefficient across thresholds was calculated.

For local accuracy measurements, the density in the 2.2 Å map was around 1.8.–3.5 Å, and individual nucleotides were used as the data using a $3\sigma$ threshold for classification. Owing to the variance in experimental uncertainty in different regions of the RNA, the AU-PRC was normalized to compare the accuracy of the molecular dynamics predictions across regions. A minimum–maximum normalization with the AU-PRC of the 2.3 Å map, AU-PRC$^{exp}$, was the ceiling (1), and the AU-PRC of the randomly shuffled densities, AU-PRC$^{random}$, was the floor (0). Any nucleotide with high experimental uncertainty (AU-PRC$^{exp} < 0.2$) was assigned a score of 0.

## Reporting summary

Further information on research design is available in the Nature Portfolio Reporting Summary linked to this article.

## Data availability

Cryo-EM maps have been deposited in the wwPDB OneDep System under the Electron Microscopy Data Bank accession codes EMD-42499 and EMD-42498 for the 2.2 Å and 2.3 Å maps, respectively. The atomic

models associated with the 2.2 Å map have been deposited in the PDB under accession codes 9CBU for the models with only the consensus waters and ions, and 9CBX for the model with all automatically identified waters and ions. The atomic models associated with the 2.3 Å map have been deposited in the PDB under accession codes 9CBW for the models with only the consensus waters and ions, and 9CBY for the model with all automatically identified waters and ions. The cryo-EM raw videos and particle stacks have been deposited to the Electron Microscopy Public Image Archive under the accession code EMPIAR-11844. Simulations can be found at the Stanford Digital Repository (https://doi.org/10.25740/sw275qs6749).

## Code availability

SWIM can be found on GitHub (https://github.com/gregdp/segger). Input files and analysis code for the molecular dynamics (including the in-house parameter file for using nMg and mMg in AMBER), and all other analysis and graphical code and modified scripts for SWIM are available on GitHub (https://github.com/DasLab/Water-CryoEM-ribozyme).

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

**Acknowledgements** This work was supported by the US National Institutes of Health (R01GM079429 to W.C. and R35GM122579 to R.D.), the National Science Foundation (2330652 to W.C. and R.D.), Bio-X Bowes Graduate Student Fellowship (to R.C.K.), the Howard Hughes Medical Institute (to R.D.), the National Key R&D Program of China (2022YFC2303700 to K.Z. and S.L., and 2022YFA1302700 to K.Z.), the Strategic Priority Research Program of the Chinese Academy of Sciences (XDB0490000 to K.Z.), the Center for Advanced Interdisciplinary Science and Biomedicine of IHM (QYPY20220019 to K.Z.), the National Natural Science Foundation of China (32301044 and 32471301 to S.L. and 32371345 to K.Z.), the Fundamental Research Funds for the Central Universities (YD9100002048 to K.Z. and YD9100002044 to S.L.) and the Cryo-EM Center at the University of Science and Technology of China (to K.Z. and S.L.). We accessed the Titan Krios electron microscope through the Stanford-SLAC Cryo-EM Center supported by the National Institutes of Health Common Fund Transformative High-Resolution Cryo-Electron Microscopy program (U24GM129541) to demonstrate the merit of using cryo-EM for RNA-only molecule high-resolution structure determination. This article is subject to HHMI's Open Access to Publications policy. HHMI laboratory heads have previously granted a non-exclusive CC BY 4.0 license to the public and a sub-licensable license to the HHMI in their research articles. Pursuant to those licenses, the author-accepted manuscript of this article can be made freely available under a CC BY 4.0 license immediately upon publication.

**Author contributions** R.C.K., G.P., K.Z., R.D. and W.C. conceived the study and designed the experiments. M.Z.P. prepared the RNA samples. K.Z. and S.L. performed the cryo-EM sample preparation, screening, data collection, image processing and initial structure modelling. G.P. implemented the updated SWIM method to detect the water and Mg²⁺ ions and refined the models. R.C.K. performed the molecular dynamics with advice from D.A.C., and analysed data with advice from R.D. R.C.K., G.P., S.L., R.D., K.Z. and W.C. wrote and edited the manuscript with input from all other authors.

**Competing interests** The authors declare no competing interests.

**Additional information**
**Correspondence and requests for materials** should be addressed to Rhiju Das, Kaiming Zhang or Wah Chiu.

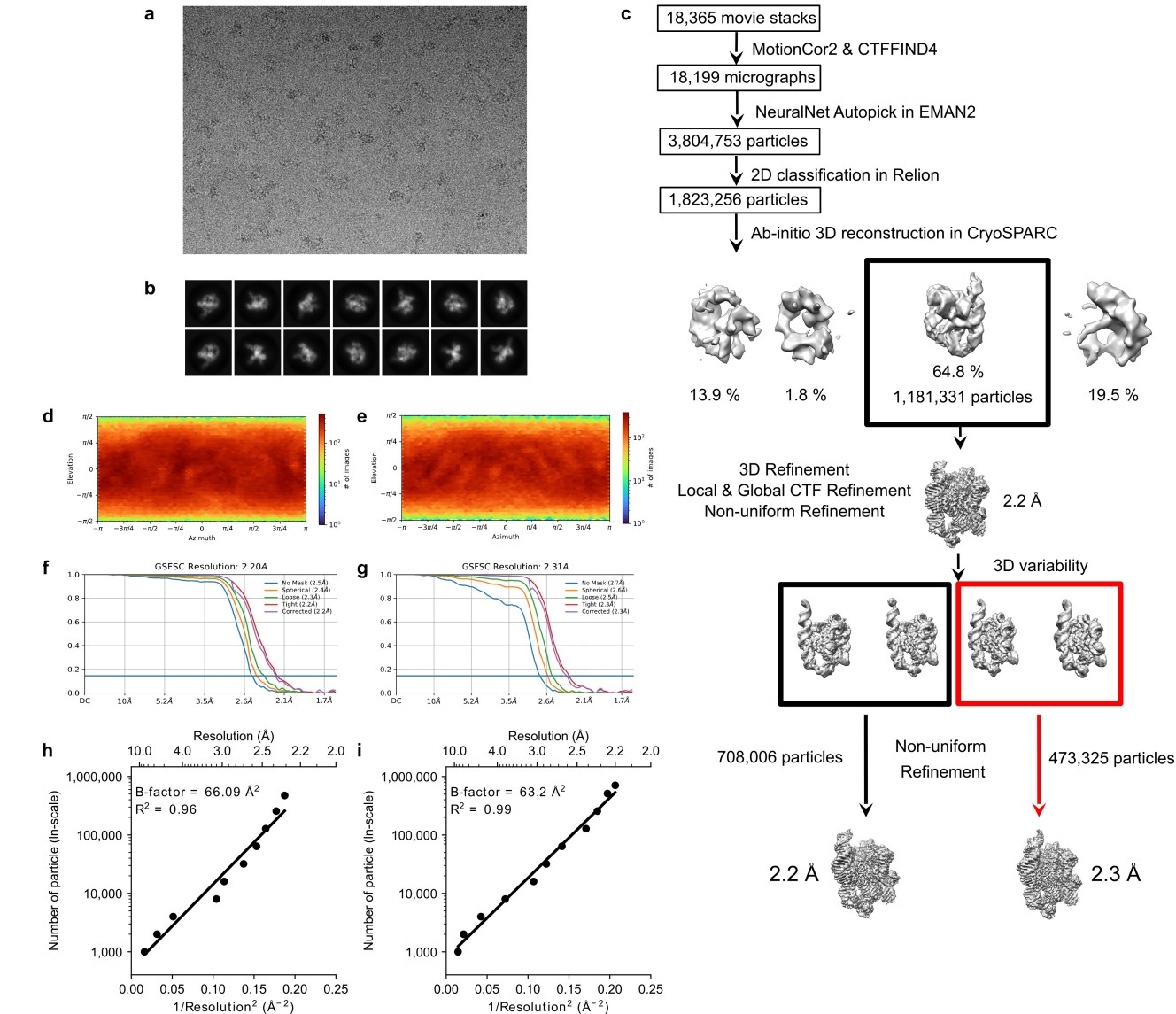

**Extended Data Fig. 1 | Workflow and quality of single-particle cryo-EM analysis of *Tetrahymena* ribozyme. (a)** Representative motion-corrected cryo-EM micrograph. **(b)** Reference-free 2D class averages of computationally extracted particles. **(c)** Workflow of image processing. **(d-e)** Euler angle distribution of the particle images for **(d)** 2.2 and **(e)** 2.3 Å maps. **(f-g)** Gold standard FSC plots for the final 3D reconstruction, calculated in CryoSPARC for **(f)** 2.2 Å and **(g)** 2.3 Å maps. **(h-i)** Plots of particle number against the reciprocal squared resolution for **(h)** 2.2 Å and **(i)** 2.3 Å maps. The B-factor was calculated as twice the linearly fitted slope and the $R^2$ value is reported.

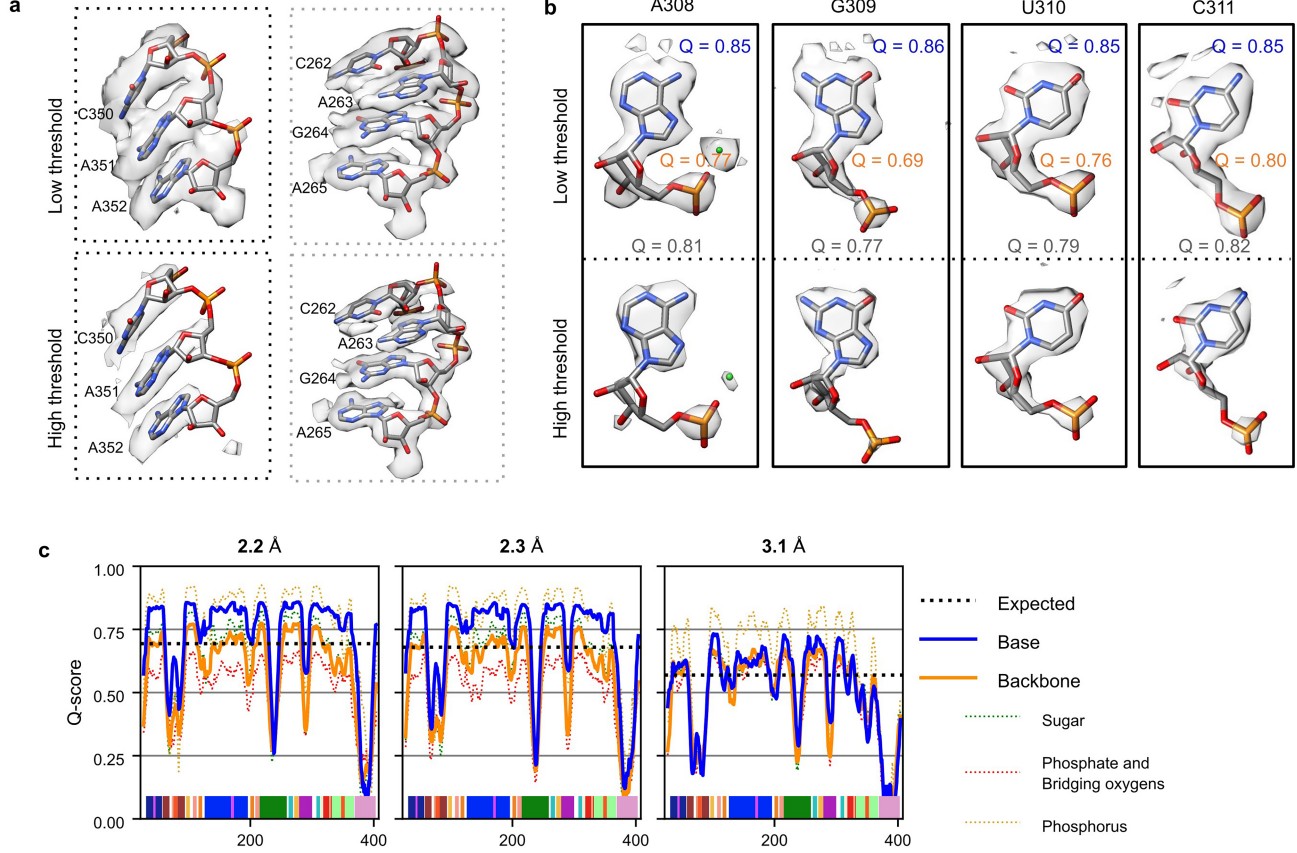

**Extended Data Fig. 2 | Structure quality of the *Tetrahymena* ribozyme cryo-EM maps.** (**a**) Visual inspection of the structure quality of the 2.2 Å map at different contour levels. (**b**) Four representative nucleotides extracted from the 2.2 Å map with high Q-scores. Each nucleotide is displayed at two density contour levels. The Q-scores for each nucleotide (gray), the backbone (orange), and the base (blue) are shown. The directional orientation of exocyclic nitrogen and oxygen densities pointing out of the bases allowed for differentiation of adenine and guanine. However, distinguishing cytosine and uracil remained difficult because their only difference lies in the exocyclic N4 (cytosine) or O4 (uracil) position. Unambiguously distinguishing between cytosine and uracil may require a resolution closer to 1 Å; for example, the difference between oxygen and nitrogen atoms becomes clear in the 1.34 Å cryo-EM map of the protein apoferritin[17]. When visually examining each nucleotide, densities of the backbone were less pronounced than the base. (**c**) Plots of per-nucleotide Q-score (rolling 10 nt average) separated by part of the nucleotide, with base in blue and the backbone in orange. The backbone was further split into the ribose sugar (green), oxygens attached to the phosphorus (pink), and phosphorus (yellow) represented by dotted lines. Nucleotide numbering, x-axis, is colored by domain matching Fig. 1. The Q-score confirmed quantitatively that resolvability is better for the bases than the backbone across the majority of the 2.2 and 2.3 Å maps. In a previous 3.1 Å map (EMDB-31385, PDB: 7ez0)[29], the base and backbone were equally resolved, indicating that increasing the resolution to ~2.3 Å is particularly important to accurately model base orientations and hence increased confidence in modeling water interactions.

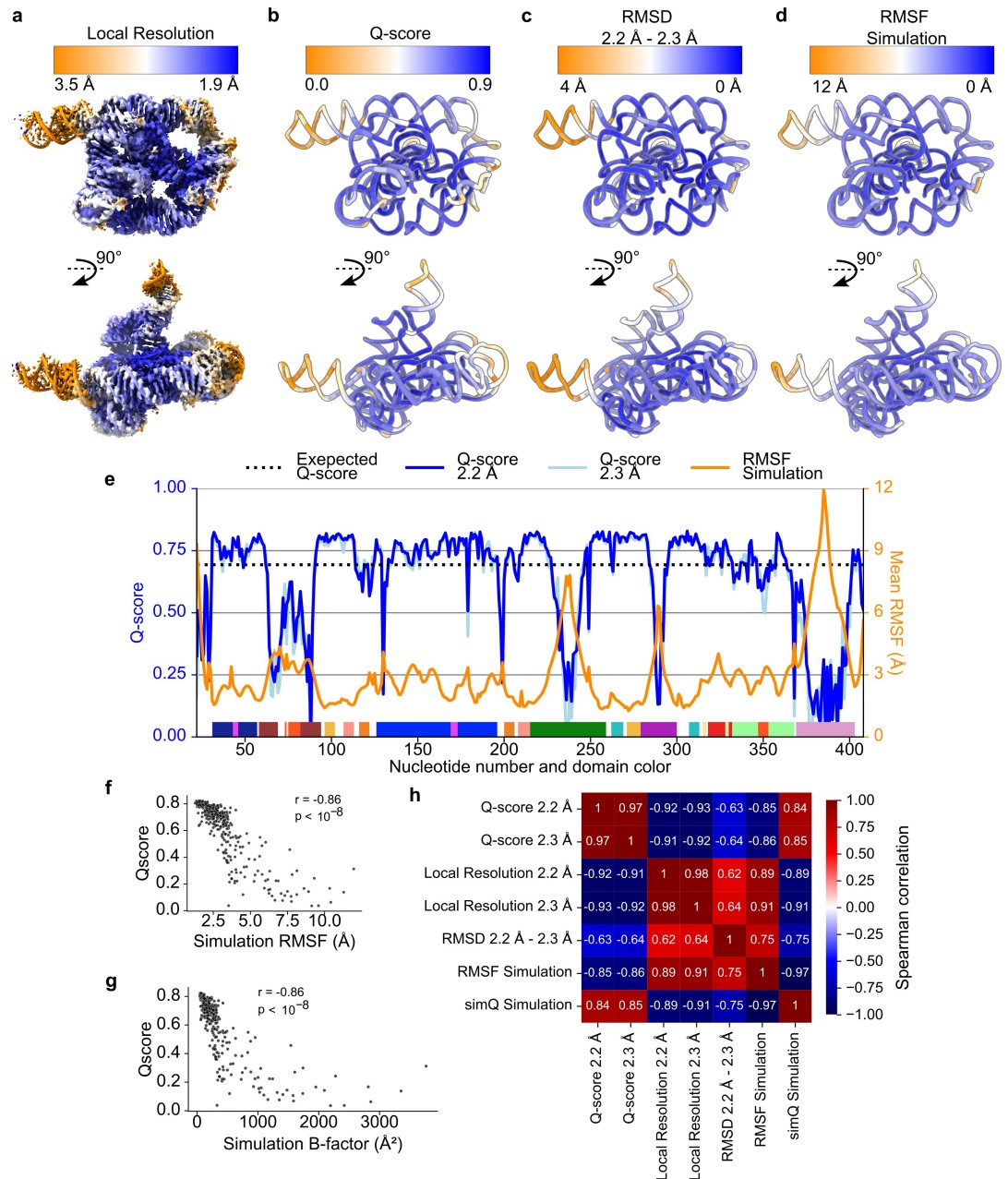

**Extended Data Fig. 3 | Domain level variability in map quality and molecular flexibility.** (**a**) Local resolution as calculated by cryoSPARC is colored from high local resolution (blue) to low local resolution (orange) on the sharpened 2.2 Å map at 3σ threshold. (**b**) The 2.2 Å model is colored by regions of high Q-score (blue) and low Q-score (orange). Several factors may contribute to the variation in local resolution and atomic resolvability, including specimen preservation, radiation damage, inherent flexibility, charge distribution, and spatial geometry. Due to the repeated structural arrangement and relatively uniform solvent accessible of RNA, we hypothesized that inherent flexibility was the major source of variation in local resolution and atomic resolvability across the ribozyme. (**c-d**) Flexibility of the structure with flexible regions colored orange, as indicated by (**c**) Root Mean Square Deviation (RMSD) of all heavy atoms between the 2.2 and 2.3 Å models, and (**d**) Root Mean Square Fluctuation (RMSF) of all heavy atoms estimated from 30 400 ns MD simulations. (**e**) Plot of per-residue Q-score and RMSF with domain colors matching Fig. 1.

(**f-g**) For each nucleotide, (**f**) the RMSF of the simulations and (**g**) the B-factor of the simulations ($\frac{8\pi^2}{3}$RMSF$^2$) of the nucleotide in MD simulation is plotted against the average Q-score of the nucleotide in the 2.2 and 2.3 Å model. Spearman's rank correlation coefficient is reported, with a strong statistically significant negative correlation, calculated with a two-sided hypothesis test for no ordinal rank between the variables ($P = 1.1 \times 10^{-116}$), supporting the hypothesis that inherent flexibility is the primary attribute explaining poor resolvability in peripheral regions of the ribozyme. The 2.2 and 2.3 Å models differed most in these poorly resolved, highly flexible regions; the classification scheme used in the image processing successfully distinguished the differences in the highly flexible regions of the RNA. (**h**) The Spearman's rank correlation coefficient was calculated between pairs of per-residue values, including the additional values of average local resolution value at the nucleotide position, RMSD between the 2.2 and 2.3 Å models, and the Q-score of the average structure from simulation in a simulation density map (simQ).

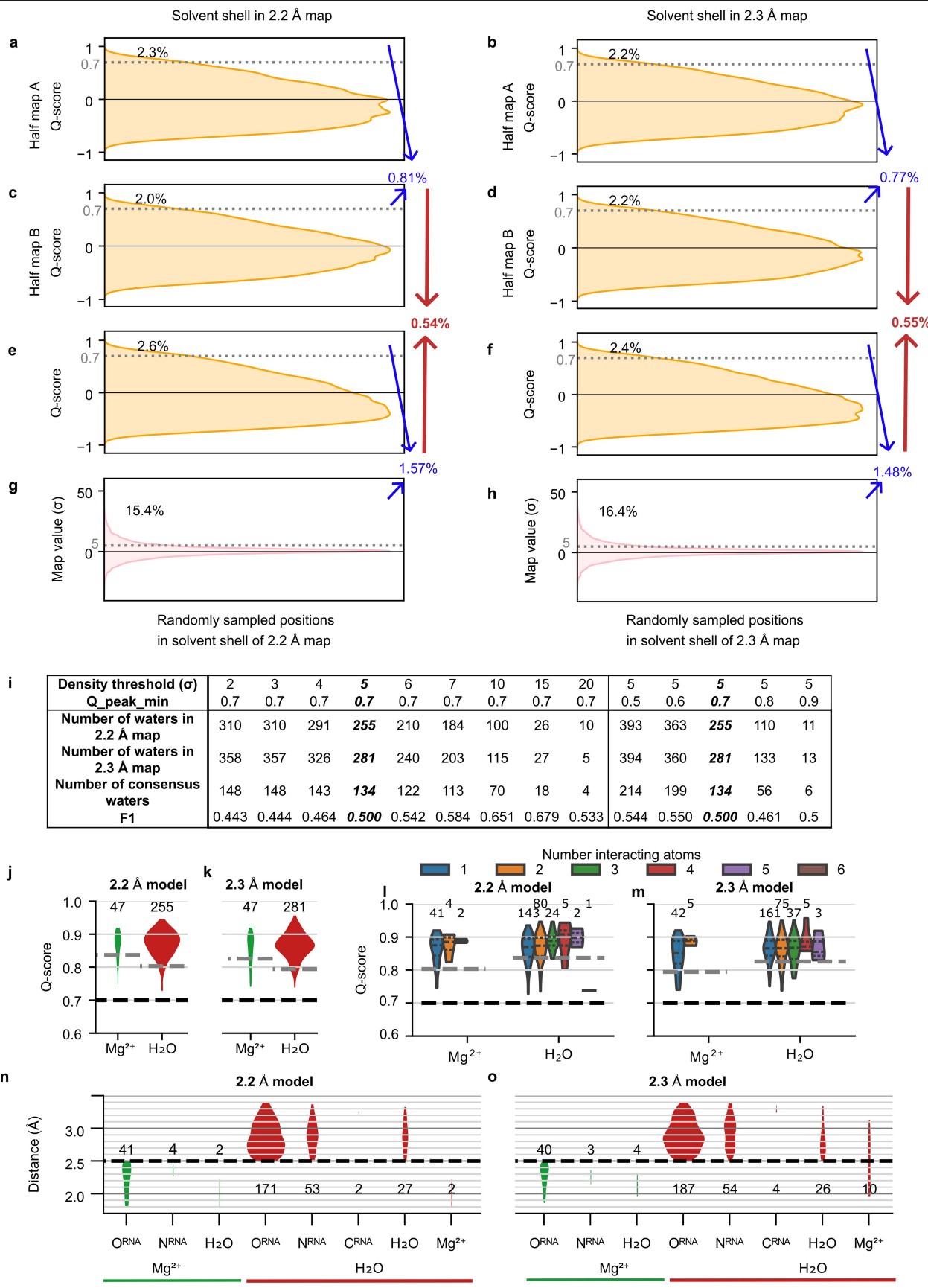

**Extended Data Fig. 4** | See next page for caption.

**Extended Data Fig. 4 | Assessment of SWIM criteria. (a-h)** Assessment of the SWIM criteria on ~89,000 randomly sampled positions in the solvent shell (1.8–3.5 Å away from RNA that has a Q-score > 0.6 indicating high reliability of the RNA model) in the **(a,c,e,g)** 2.2 and **(b,d,f,h)** 2.3 Å maps. The distribution of **(a-d)** Q-score in each half-map, **(e-f)** Q-score in full map, and **(g-h)** map density-value are plotted. Cutoff values used for SWIM in this study are labeled with a gray dotted line (0.70 Q-score in full and half maps and 5σ above map mean value). The percent of randomly sampled solvent shell positions above these thresholds are labeled black. Then combining the criteria, the red number reports the number of positions that pass all the SWIM criteria. Our analysis showed that the SWIM criteria are stringent; only 0.6% of positions in the solvent shell of the maps pass all SWIM criteria. **(i)** The SWIM criteria are systematically varied, the number of waters identified in each map, the number of consensus waters between the two-map and the F1-score between the set of waters identified in the two maps are reported. **(j-k)** Distribution of Q-scores for SWIM-modeled water and $Mg^{2+}$ ions. Expected Q-score at resolution for water and ion labeled as gray dashed line and the SWIM imposed Q-score threshold is labeled as a black dashed line. **(l-m)** As in J-K, but the water and $Mg^{2+}$ ions are split by the number of interacting (<3.5 Å) RNA atoms. A trend shows an increase in Q-score with more interacting atoms but the trend is not statistically significant (linear regression, two-sided t = 1.826, p = 0.067). **(n-o)** Distribution of distances between water and $Mg^{2+}$ ion and nearest atom. Thresholds used by SWIM are labeled as a black dashed line.

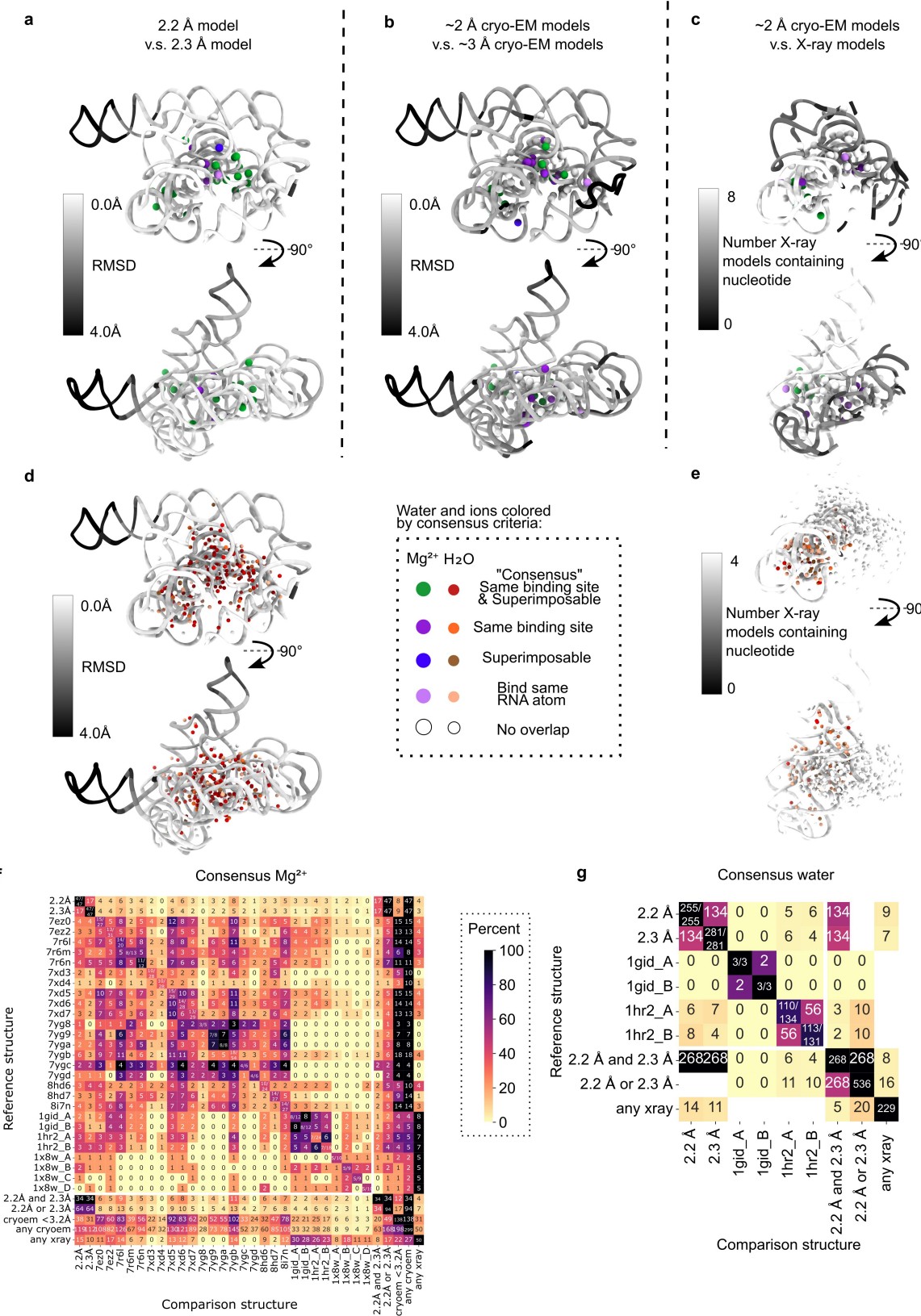

**Extended Data Fig. 5** | See next page for caption.

**Extended Data Fig. 5 | Comparison of SWIM-modeled water and Mg$^{2+}$ ions to previous models.** (**a-e**) Comparing the waters and Mg$^{2+}$ ions modeled in (**a,d**) the 2.2 Å model to the 2.3 Å model, (**b**) the 2.2 and 2.3 Å models to ~3 Å cryo-EM models[29,33,34,85], and (**c,e**) the 2.2 and 2.3 Å models to X-ray models[8,28,32]. Water and Mg$^{2+}$ ions that bind the same RNA atoms (criterion 1) and are within 1 Å after local superposition (criterion 2) are red and green, respectively; only meet criterion 1 are orange and purple, respectively; those that only meet the criterion 2 are brown and blue, respectively; those that bind one of the same RNA atoms are light orange and light purple, respectively and those with no overlap are white. RNA is colored by model overlap as evaluated by (**a,b,d**) average RMSD between structures and (**c,e**) the number of models that contain each nucleotide (since the X-ray structures are not the full length ribozyme). (**f-g**) Consensus of (**f**) Mg$^{2+}$ ion and (**g**) water from the structures in this study and previous structures. The diagonal of the plot notes how many water or Mg$^{2+}$ ions pass the current distance criteria used in this study (2.5 Å and 3.5 Å respectively), only these Mg$^{2+}$ ions and water are compared. Note that although other structures did model Mg$^{2+}$ ions that bind RNA in their second coordination shell (through a water), these were not considered. On the off diagonal the number of water or Mg$^{2+}$ ions bound to the same RNA atoms is reported and colored by the percent of the total water or Mg$^{2+}$ ions from the reference structure on that row. The final rows and columns of (**F**) compare to the 34 Mg$^{2+}$ ions in both the 2.2 or 2.3 Å models, 94 Mg$^{2+}$ ions in either the 2.2 or 2.3 Å models, the 138 Mg$^{2+}$ any previous cryo-EM model with resolution better that 3.2 Å, the 299 Mg$^{2+}$ in any previous cryo-EM models, and the 50 Mg$^{2+}$ found in any X-ray models. The final rows and columns of (**G**) compare to the 268 waters in both the 2.2 and 2.3 Å models, 536 waters in either the 2.2 and 2.3 Å models, and the 229 waters found in any X-ray models.

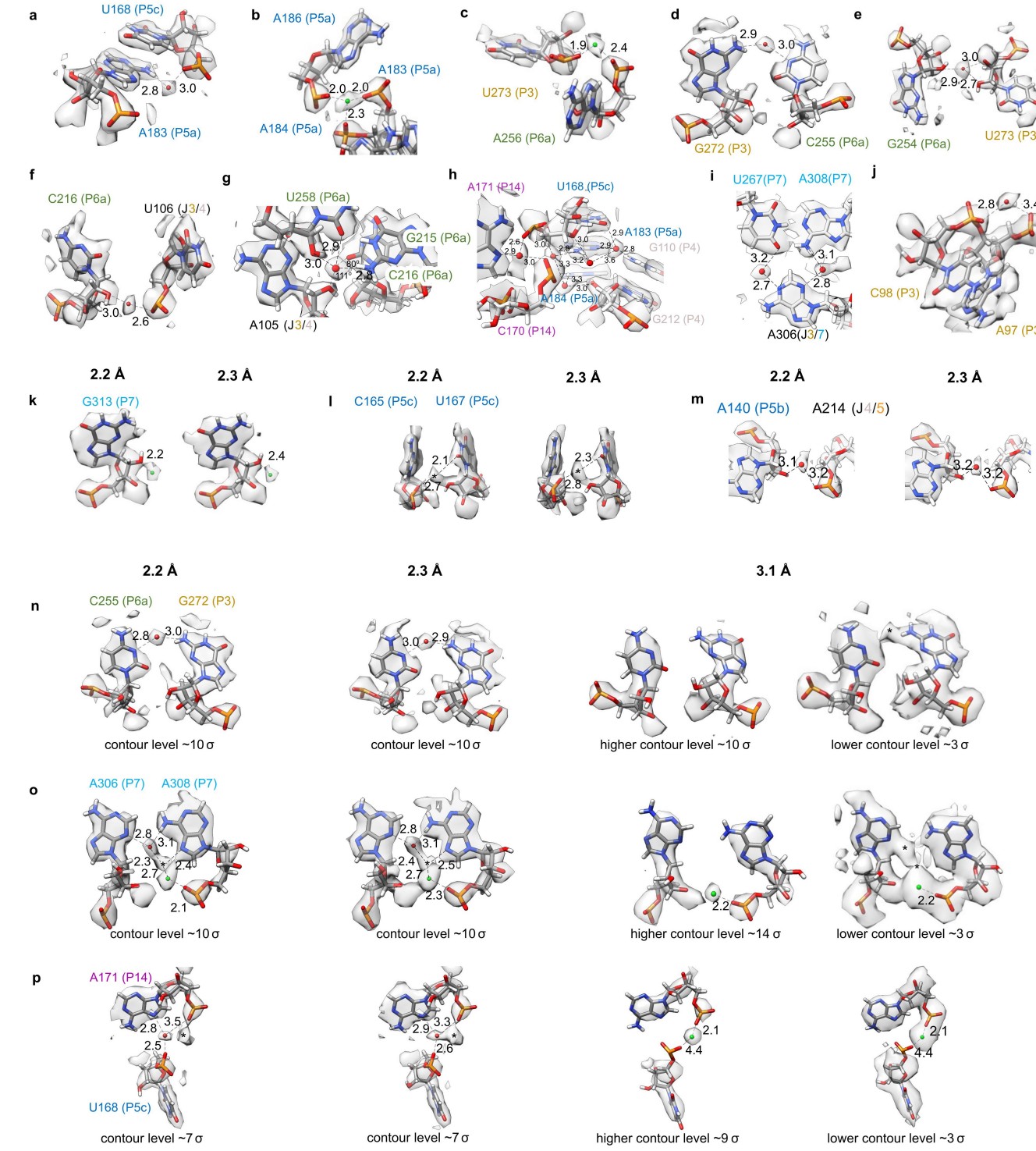

**Extended Data Fig. 6** | See next page for caption.

**Extended Data Fig. 6 | Water and Mg$^{2+}$ ion binding sites.** Water (red spheres) and Mg$^{2+}$ ions (green spheres) found in both cryo-EM maps are displayed. Only some of the contacts for each water molecule and Mg$^{2+}$ ion may be shown in each case. (**a-j**) highlight a selection of regions from the 2.3 Å map and model. The identical regions but in the 2.2 Å map and model can be found in Fig. 3. (**a**) Mg$^{2+}$ ion found in previous X-ray and cryo-EM structures, which binds the phosphates of U273 and A256. (**b**) A water anchors the base of A183 in the P5a domain to the P5c domain by bridging the amine (N6) of A183 to the phosphate of U168. This water was previously modeled in chain B of the crystal structure 1hr2[8]. (**c**) The backbone of A183 is further involved in the previously described metal ion core[32] in the A-rich bulge of domain P5a. The second Mg$^{2+}$ ion of the metal ion core was also modeled in both maps (not shown). Mg$^{2+}$ ions are integral to folding of the P4-P6 domain, shielding the negative charge of the phosphate backbone[32]. (**k-m**) There were additional peaks, present in both maps, that raised more ambiguity in assignment, particularly Mg$^{2+}$ ions bound to sugars. (**k**) SWIM modeled a density peak 2.2/2.4 Å away from the 2′ hydroxyl group of the G313 sugar which was too close for water so, due to limitations mentioned previously, we have modeled it as a Mg$^{2+}$ ion, although the identity could be a different ion. (**l**) SWIM identified a peak in both maps that is 2.1/2.3 Å away from a 2′ hydroxyl suggesting a Mg$^{2+}$ ion, but 2.7/2.8 Å away from a phosphate, suggesting a water. SWIM would have modeled a Mg$^{2+}$ ion, but this position was too near to another Mg$^{2+}$ ion, so this position was left unmodeled. This position was modeled as a water in a crystal structure of the P4-P6 domain (PDB: 1hr2)[8]. (**m**) Elsewhere in the model, a peak is observed in a similar chemical environment to (**l**) although the longer bond length clearly identifies this peak as water. This water displayed how, even within a region where nucleotides do not interact directly, water bridges the backbone of nucleotides to stabilize the RNA fold which brings these nucleotides in close proximity. Finally, (**n-p**) show the same regions in the 2.2 Å, 2.3 Å, and 3.1 Å (EMDB-31385, PDB: 7ez0)[29] maps with contour noted below each panel and possible binding sites are marked with a *.

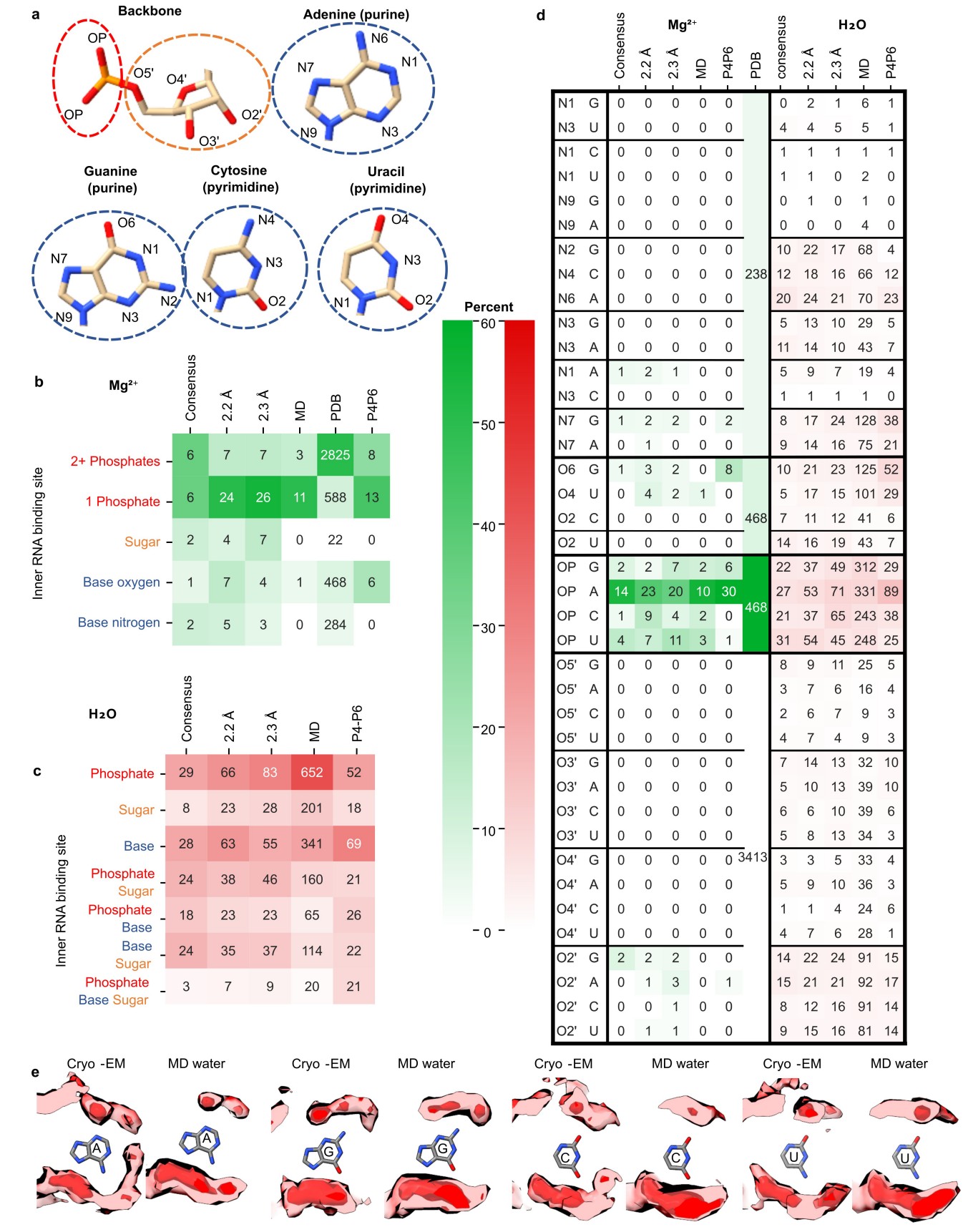

**Extended Data Fig. 7 |** See next page for caption.

**Extended Data Fig. 7 | Analysis of water and ion RNA binding motifs.** To obtain an overview of chemical binding motifs identified in cryo-EM maps of RNA in vitreous ice, we compared the prevalence of binding motifs of water and Mg[2+] ions from the consensus waters and ions between the two cryo-EM maps presented here, the MD simulations, and a published curated set of 489 X-ray PDB structures with 15,334 Mg[2+] ions called MgRNA[38]. (**a**) Diagram of atoms in RNA backbone and nucleotides. (**b-c**) The frequency of binding motifs for (**b**) Mg[2+] ions and (**c**) water defined as RNA atoms within a 2.5 and 3.5 Å distance, respectively. The cryo-EM counts are labeled for consensus and the maps individually, the number of unique Mg[2+] and water binding sites in MD, for Mg[2+], the binding counts in the curated set of PDB Structures MgRNA[38], and finally the binding site count in just P4P6 (PDB 1HR2 and 1GID[8,32]). It is colored by percent frequency of that interaction type. (**d**) Same as (b-c) but now separated by every atom type. Our cryo-EM structures bind single phosphate more frequently than multiple phosphates whereas the PDB contains more Mg[2+] ions bound to multiple phosphates. But when limiting the PDB dataset to just the P4-P6 subdomain of the ribozyme, there are similar binding preferences between cryo-EM, MD, and previous experimental results[15,86], indicating that motif frequencies are dependent on the RNA probed and are consistent across techniques for the *Tetrahymena* ribozyme. (**e**) In red is the average cryo-EM density (>1.8 Å from modeled RNA) and the average MD density for the same residues as calculated from 30 400 ns simulations for every A, U, G, and C in the ribozyme respectively.

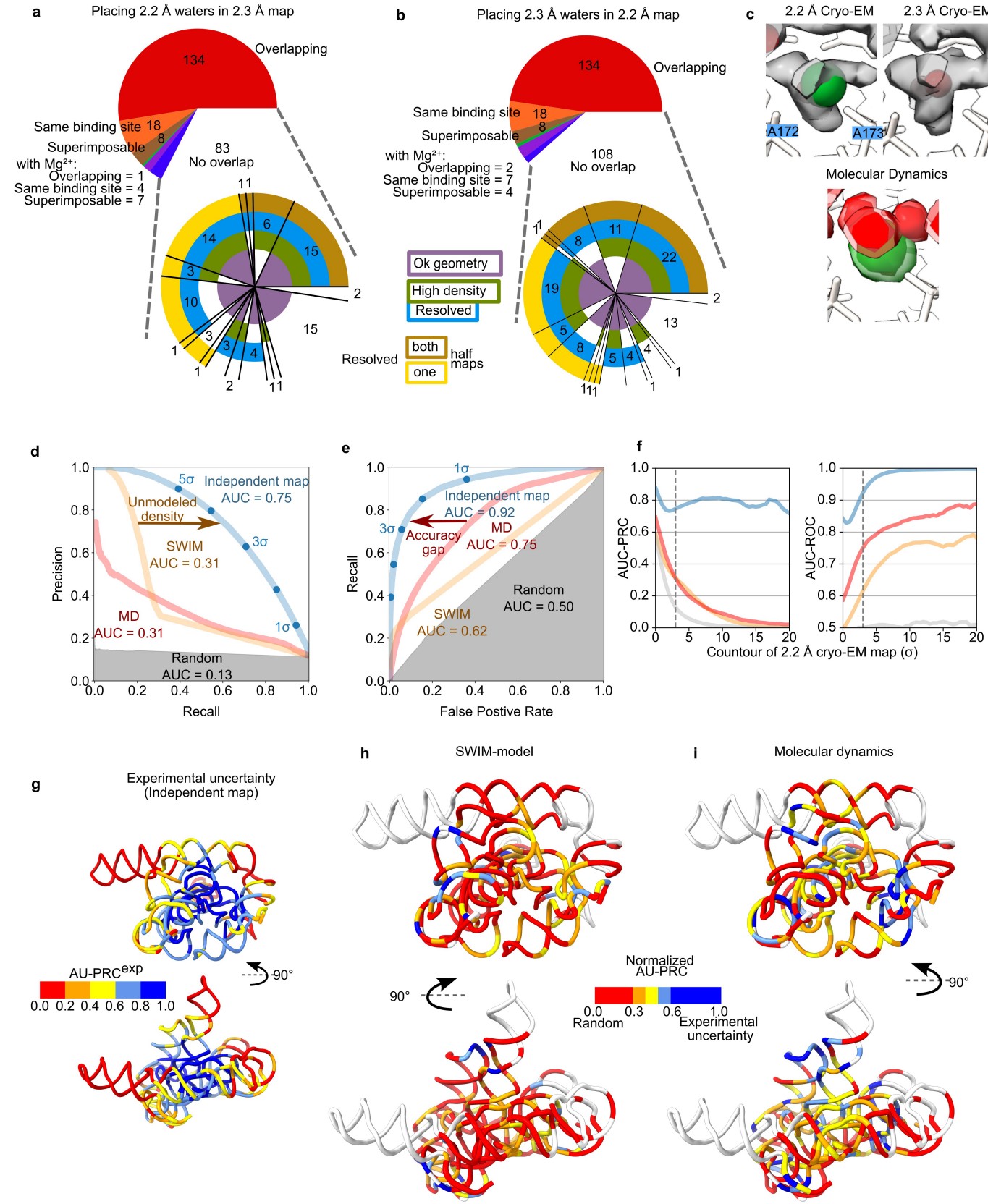

**Extended Data Fig. 8** | See next page for caption.

**Extended Data Fig. 8 | Analysis of overlap solvent shell density.**
(**a-b**) Description of how SWIM-assigned waters (**a**) from the 2.2 Å model match the 2.3 Å model and map and (**b**) water from the 2.3 Å model match the 2.2 Å model and map. The top pie chart describes how the waters overlap with waters or ions in the other model after a local superposition, with red indicating overlap, same binding site and superimposable, with water in the other model, orange indicating the same binding site only and brown indicating superimposable only (Methods). Likewise for $Mg^{2+}$ ions in the other model, the same categories are colored green, purple, blue respectively. For the waters with no overlap in the model, the bottom pie chart describes whether the water when placed in the other map has a geometry (purple), density value (> 5σ, green), and/or resolvability (Q-score > 0.70, blue, brown, yellow for full, both half maps, one half map respectively) that pass the SWIM criteria. (**c**) A specific example of how a $Mg^{2+}$ ion can overlap with a water displayed as in Fig. 5. The cryo-EM densities have the star-shaped shape indicative of a $Mg^{2+}$ ion with a full-coordination shell. The MD agrees that this site contains a $Mg^{2+}$ ion coordinated to the two phosphate and 4 waters. Due to the blurring of density automated SWIM identified two different peak positions within the diffuse density, and modeled a $Mg^{2+}$ ion in the 2.2 Å map and a water in the 2.3 Å map. (**d–i**) The agreement between the reference map, the 2.2 Å cryo-EM map, and the comparison maps, 2.3 Å cryo-EM map (blue), 2.2 Å model (orange; created by molmap), the MD water density (red), and the 2.2 Å cryo-EM map with the voxel values shuffled (gray). The blue line represents experimental uncertainty while the gray represents performance of a random algorithm. Voxels are defined as "positive" if they have a value above the given contour, 3σ for (**d-e**), in the reference map and "negative" otherwise. Other metrics are tabulated in Supplementary Table 2. (**d-e**) The contour of the comparison map is varied to plot (**d**) Precision-recall (PRC) and (**e**) receiver operating characteristic (ROC) curves. The Area Under the Curve (AUC) is labeled. (**f**) The AUC is calculated for a variety of reference map, the 2.2 Å cryo-EM map, contours and for the solvent shell. Trends are consistent over reasonable contours of the reference map, the 2.2 Å map. (**g**) The model is colored by per-nucleotide experimental precision, defined as the AU-PRC for the independent map. This shows precision decreasing radially from the center of mass of the molecule. (**h-i**) The local agreement of (**h**) the SWIM model and (**i**) the MD density to the reference map, the 2.2 Å cryo-EM map, is plotted, normalized AU-PRC such that a score of 1 is equal to experimental uncertainty (see Methods). The normalized AU-PRC is plotted on the structure with a view of the catalytic core (top) and the tetraloop receptor (bottom), from low agreement (red) to high agreement (blue). Nucleotides with water structure with high experimental uncertainty (AU-PRC$^{exp}$ < 0.2) are white.

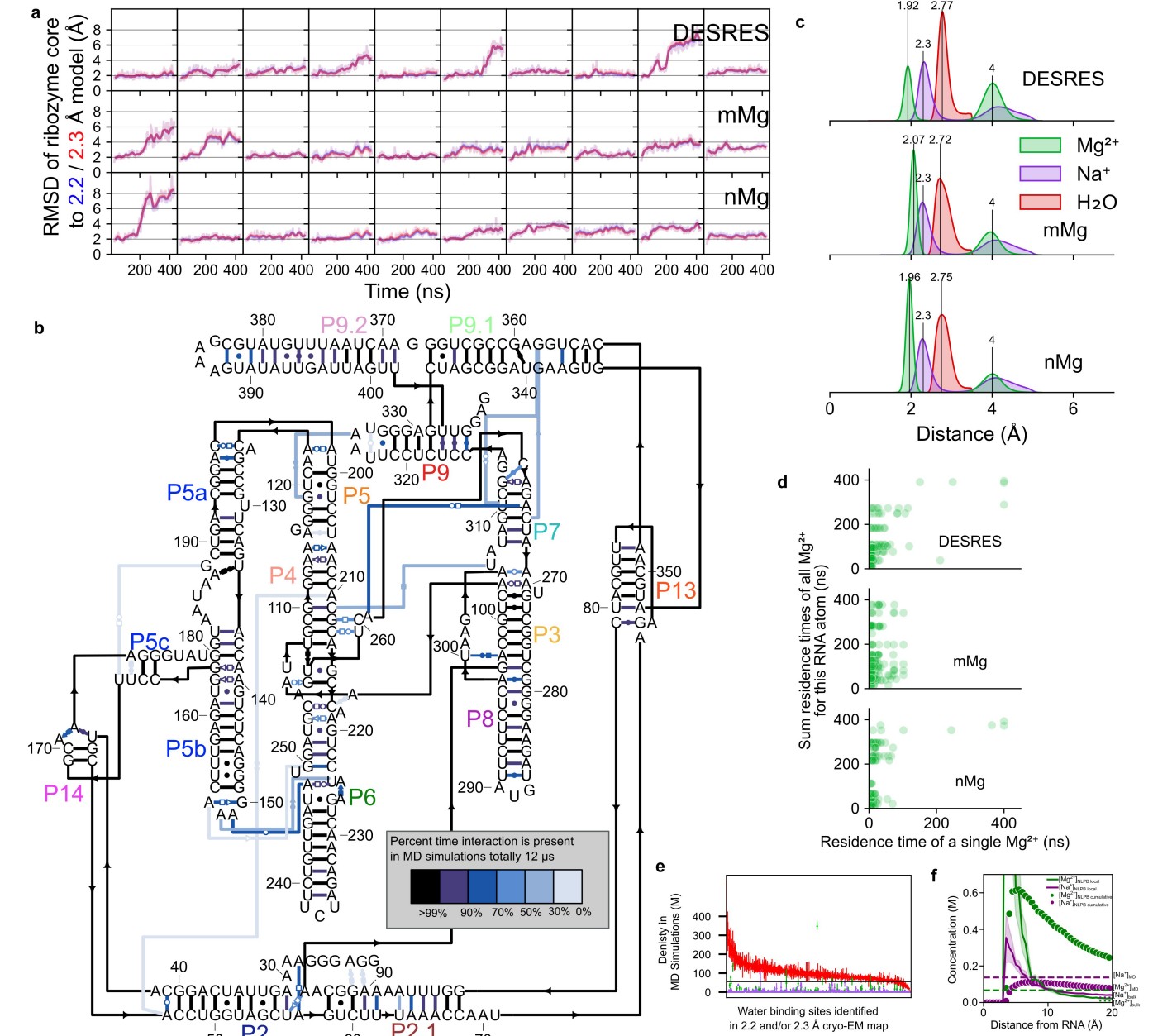

**Extended Data Fig. 9** | See next page for caption.

**Extended Data Fig. 9 | Molecular dynamics simulation stability and water and ion characteristics.** (**a**) Root-mean-square deviation (RMSD) of snapshots (1 ns) of the MD simulations from the initial 2.2 Å (blue) model and the 2.3 Å (red) model in solid and the 20 ns moving average in light. The top row is from 10 simulations using the DESRES forcefield, the second and third row using the parmBSC0χOL3 force field with the mMg and nMg $Mg^{2+}$ parameters respectively. The first six columns have extra ions initially randomly placed, the last four columns have extra ions initially placed electrostatically. The simulations were stable as measured by the high similarity to both the cryo-EM structures throughout the 400 ns simulation. The simulations were not biased towards the simulation starting conformation, 2.2 Å model, as seen by the overlap of red and blue lines. Therefore, the simulations were used to interpret both the 2.2 and 2.3 Å cryo-EM structures. (**b**) Secondary structure diagram of the starting coordinates with each starting interaction colored by what percent of time, through all simulations, the interaction is present. All simulations maintained their tertiary structure except for three (10%) simulations where the P9-P5 interaction broke and P9 shifted away from the core; frames with these high deviations to cryo-EM structures were excluded from further analysis. The MD simulations described above were conducted in explicit solvent and included $Mg^{2+}$ and $Na^+$ ions. (**c**) Density plot of minimum distance between $Mg^{2+}$ ions (green), $Na^+$ ions (purple), or waters (red) and RNA in all molecular dynamics simulations split by force field used (for each force field, 400 ns simulations, N = 10). Waters past 3.5 Å were not measured. The water and $Mg^{2+}$ ions exhibited expected bond lengths to RNA; water peaked at 2.7–2.8 Å while $Mg^{2+}$ ions peaked between 1.9 and 2.1 Å in the first shell and at ~4 Å in a second coordination shell. The force fields used differed most in their parameterization of $Mg^{2+}$ ions where mMg and nMg force fields were designed to enable faster $Mg^{2+}$ ion exchange to increase sampling of $Mg^{2+}$ binding[58,79]. (**d**) For each MD $Mg^{2+}$ ion binding site which is bound to a phosphate, the residence time of a single binding event is plotted against the total occupancy of $Mg^{2+}$ ions at that binding site. The simulations are separated by force field (rows). We affirmed the difference between force fields; $Mg^{2+}$ ions resided for shorter times when bound to phosphates and carbonyls of the RNA in the mMg and nMg force fields than the DESRES force field where $Mg^{2+}$ ions bound to phosphates rarely unbind in the 400 ns time of simulation. (**e**) The mean water density in the MD simulations (red) is plotted for each water binding site found in the 2.2 or 2.3 Å maps. The highest density within 1 Å of the cryo-EM coordinate was reported. A density value of 55 M is equivalent to the density of bulk water (black line). The minimum and maximum density value between the three force fields used is plotted as error bars. The mean densities, and range of densities across force fields, of $Mg^{2+}$ ions (green) and $Na^+$ ions (purple) ions are also plotted for these binding sites. (**f**) Comparison of concentration of $Mg^{2+}$ ions (green) and $Na^+$ ions (purple) from simulation (dotted horizontal lines) and expected local concentration (solid line) based on non-linear Poisson-Boltzmann theory (NLPB) with APBS[87] using the bulk concentration used in the experiment (dotted horizontal line on right of plot). The shaded error bar represents the 95% confidence interval. The concentration of $Na^+$ ions used in MD simulation is slightly greater than the expected cumulative concentration of $Na^+$ ions from NLPB theory (dotted curve), which is itself known to overpredict bound $Na^+$ ions relative to competing $Mg^{2+}$ ions in nucleic acids' ion atmospheres. This observation suggests that MD simulations are unlikely to be underestimating the density of $Na^+$ ions bound to RNA. Combined with the observation of only a handful of $Na^+$ binding sites in MD simulations, this analysis suggests that very few of the cryo-EM density features surrounding the ribozyme should be attributed to $Na^+$ ions. This analysis supports our approximation that those map features are either water or $Mg^{2+}$ ions, which greatly outnumber $Na^+$ ions.

**Extended Data Table 1 | Cryo-EM data collection, refinement, and summary validation statistics**

| | *Tetrahymena* ribozyme 2.2 Å (EMDB-42499) (PDB 9CBU, 9CBX) | *Tetrahymena* ribozyme 2.3 Å (EMDB-42498) (PDB 9CBW, 9CBY) |
|---|---|---|
| **Data collection and processing** | (EMPIAR 11844) | |
| Magnification | 105,000 | |
| Voltage (kV) | 300 | |
| Microscope | Titan Krios G3 | |
| Detector | Gatan K3 | |
| Energy filter slit width (eV) | 20 | |
| C2 aperture size (μm) | 70 | |
| Objective aperture size (μm) | 100 | |
| Electron exposure (e⁻/Å²) | 57.3 | |
| Exposure time (s) | 2.5 | |
| Defocus range (μm) | -0.5 to -2.0 | |
| Pixel size (Å) | 0.82 | |
| Symmetry imposed | C1 | |
| Number of micrographs collected | 18,365 | |
| Number of micrographs used | 18,199 | |
| Initial particle images (no.) | 3,804,753 | |
| Final particle images (no.) | 708,006 | 473,325 |
| Map resolution (Å) | 2.2 | 2.3 |
| FSC threshold (0.143) | | |
| Map resolution range (Å) | 2.0-5.6 | 2.1-5.8 |
| Average Q-score RNA | 0.649 | 0.638 |
| Average Q-score H₂O | 0.863 | 0.848 |
| Average Q-score Mg²⁺ | 0.868 | 0.863 |
| **Refinement** | | |
| Initial model used (PDB code) | 7ez0 | |
| Refinement programs used | Phenix, ISOLDE | |
| Map sharpening $B$ factor (Å²) | 79.23 | 74.07 |
| R.m.s. deviations | | |
| Bond lengths (Å) | 0.0136 | 0.0136 |
| Bond angles (°) | 1.85 | 1.84 |
| **Validation** | | |
| MolProbity score | 1.74 | 1.71 |
| Clashscore | 0.24 | 0.16 |

2024-06-11682

# Reporting Summary

## Statistics

For all statistical analyses, confirm that the following items are present in the figure legend, table legend, main text, or Methods section.

| n/a | Confirmed | |
|---|---|---|
| ☒ | ☐ | The exact sample size (*n*) for each experimental group/condition, given as a discrete number and unit of measurement |
| ☒ | ☐ | A statement on whether measurements were taken from distinct samples or whether the same sample was measured repeatedly |
| ☐ | ☒ | The statistical test(s) used AND whether they are one- or two-sided *Only common tests should be described solely by name; describe more complex techniques in the Methods section.* |
| ☒ | ☐ | A description of all covariates tested |
| ☒ | ☐ | A description of any assumptions or corrections, such as tests of normality and adjustment for multiple comparisons |
| ☐ | ☒ | A full description of the statistical parameters including central tendency (e.g. means) or other basic estimates (e.g. regression coefficient) AND variation (e.g. standard deviation) or associated estimates of uncertainty (e.g. confidence intervals) |
| ☐ | ☒ | For null hypothesis testing, the test statistic (e.g. *F*, *t*, *r*) with confidence intervals, effect sizes, degrees of freedom and *P* value noted *Give P values as exact values whenever suitable.* |
| ☒ | ☐ | For Bayesian analysis, information on the choice of priors and Markov chain Monte Carlo settings |
| ☒ | ☐ | For hierarchical and complex designs, identification of the appropriate level for tests and full reporting of outcomes |
| ☐ | ☒ | Estimates of effect sizes (e.g. Cohen's *d*, Pearson's *r*), indicating how they were calculated |

*Our web collection on statistics for biologists contains articles on many of the points above.*

## Software and code

Policy information about availability of computer code

| Data collection | EPU software (Thermo Fisher Scientific, version 2.7) |
|---|---|
| Data analysis | MotionCor2 (1.2.6), CTFFIND4 (4.1.12), EMAN2 (20200925), Relion (3.0), cryoSPARC (3.2.0), phenix (1.14; including the molprobity plugin), ISOLDE (ChimeraX v1.6.1), Amber (20), MDAnalysis (2.7), Q-score https://github.com/gregdp/mapq (1.9.12), Segger https://github.com/gregdp/segger (v2.9.7), manuscript scripts: https://github.com/DasLab/Water-CryoEM-ribozyme |

For manuscripts utilizing custom algorithms or software that are central to the research but not yet described in published literature, software must be made available to editors and reviewers. We strongly encourage code deposition in a community repository (e.g. GitHub). See the Nature Portfolio guidelines for submitting code & software for further information.

## Data

Policy information about availability of data

All manuscripts must include a data availability statement. This statement should provide the following information, where applicable:
- Accession codes, unique identifiers, or web links for publicly available datasets
- A description of any restrictions on data availability
- For clinical datasets or third party data, please ensure that the statement adheres to our policy

Cryo-EM maps have been deposited in the wwPDB OneDep System under EMD accession codes 42499, 42498 for the 2.2 and 2.3 Å map respectively. The atomic models associated with the 2.2 Å map are deposited in the PDB under accession codes 9CBU for the models with only the consensus waters and ions and 9CBX for

# Research involving human participants, their data, or biological material

Policy information about studies with human participants or human data. See also policy information about sex, gender (identity/presentation), and sexual orientation and race, ethnicity and racism.

| | |
|---|---|
| Reporting on sex and gender | N/A |
| Reporting on race, ethnicity, or other socially relevant groupings | N/A |
| Population characteristics | N/A |
| Recruitment | N/A |
| Ethics oversight | N/A |

Note that full information on the approval of the study protocol must also be provided in the manuscript.

# Field-specific reporting

Please select the one below that is the best fit for your research. If you are not sure, read the appropriate sections before making your selection.

☒ Life sciences ☐ Behavioural & social sciences ☐ Ecological, evolutionary & environmental sciences

For a reference copy of the document with all sections, see nature.com/documents/nr-reporting-summary-flat.pdf

# Life sciences study design

All studies must disclose on these points even when the disclosure is negative.

| | |
|---|---|
| Sample size | Sample sizes were not predetermined. The number of particles used for cryo-EM reconstruction was limited by the 48 hours of collection time, the particles were sufficient for the desired resolution. The number of magnesium ions and water ions analyzed were limited by resolution of the map, there was insufficient magnesiums for a full analysis, hence the full analysis was only conducted on water. |
| Data exclusions | In cryo-EM, particles were excluding during 2D classification and 3D reconstruction as is standard in the field. For molecular dynamics frames were excluded that deviated too far from the starting orientations, as defined by RMSD. |
| Replication | In cryo-EM the standard is not to do full replicates for reconstruction, but instead reconstruct two halves of the data independently, this was done and the cryo-EM maps were "replicable" up to 2.2 and 2.3 Å. For the molecular dynamics, 30 replicates were conducted in total, and these are reproducible for the properties analysed here-in. |
| Randomization | This is irrelevant to the study as their were no groupings. For the molecular dynamics, initial water and metal placements were randomized for some simulations. |
| Blinding | Blinding was not relevant for this study, structure of this RNA was known to all researcher involved in solving this structure. |

# Reporting for specific materials, systems and methods

We require information from authors about some types of materials, experimental systems and methods used in many studies. Here, indicate whether each material, system or method listed is relevant to your study. If you are not sure if a list item applies to your research, read the appropriate section before selecting a response.

## Materials & experimental systems

| n/a | Involved in the study |
|---|---|
| ☒ ☐ | Antibodies |
| ☒ ☐ | Eukaryotic cell lines |
| ☒ ☐ | Palaeontology and archaeology |
| ☒ ☐ | Animals and other organisms |
| ☒ ☐ | Clinical data |
| ☒ ☐ | Dual use research of concern |
| ☒ ☐ | Plants |

## Methods

| n/a | Involved in the study |
|---|---|
| ☒ ☐ | ChIP-seq |
| ☒ ☐ | Flow cytometry |
| ☒ ☐ | MRI-based neuroimaging |

## Plants

| | |
|---|---|
| Seed stocks | N/A |
| Novel plant genotypes | N/A |
| Authentication | N/A |

