## [Peer Review file · Nature]

Complex water networks visualized by cryogenic electron microscopy of RNA

Corresponding Author: Professor Wah Chiu

Version 0:

Reviewer comments:

Referee #1

(Remarks to the Author)

This study employed cryo-EM to visualize the Tetrahymena ribozyme at resolutions of 2.2 and 2.3 Å, uncovering intricate water interactions within the RNA structure. The authors used a Segmentation-guided Water and Ion Modeling (SWIM) to identify water molecules and Mg²⁺ ions from non-proteinaceous electronic density in the cryo-EM maps. This task is challenging due to the inherent ambiguity in interpreting electronic density. The paper is very well written and rigorously performed, synergistically combining molecular dynamics simulations, cryo-EM and computational models for electron density refinement. I support the publication of this article in Nature upon moderate revisions, as outlined below.

Through cryo-EM and MD simulations, the authors observed extensive mediation of RNA non-canonical interactions by water molecules, with both rigid and flexible waters. Intriguingly, regions where SWIM did not model ordered water showed similar densities in both cryo-EM maps, suggesting these regions should be assigned as water, indicating complex water networks as predicted by MD. While there is general agreement between MD simulations and cryo-EM data, the study notes that the MD-predicted water structure performs poorly in certain regions, particularly within the catalytic core. This suggests limitations in the accuracy of MD simulations in these critical areas, likely due to the constraints of the force-field models. These limitations can be addressed through quantum-classical simulations, which allow for the direct computation of electronic densities in critical regions, including active sites. By comparing and fitting these computed densities with experimental data, as demonstrated by Schwartz et al. (Nat. Commun. 15:3324 (2024)), the accuracy of simulations can be significantly improved.

The authors report that nMg ions were parameterized using an internally developed parameter file. However, the performance of this Mg model relative to the current leading models in the field remains unclear. Divalent metal ions like Mg²⁺ and Mn²⁺ present unique challenges in simulation modeling. The Li & Merz 12-6-4 models are noteworthy for accurately representing the behavior of these ions, as highlighted in their ability to replicate a wide range of bulk properties (J. Chem. Theory Comput. 10:289–97). When simulating nucleic acid-bound metals, it is essential to include specific pairwise corrections, such as those developed by Panteva and colleagues (J. Phys. Chem. B. 119:15460–70). These corrections are crucial for accurately adjusting the interactions between divalent metal ions (e.g., Mg²⁺, Mn²⁺) and water molecules, which play a significant role in the system investigated here.

The authors acknowledge that the ions modeled using SWIM only neutralize about one-quarter of the RNA charge. Do the authors find additional sites for ion binding during MD simulations?

Referee #3

(Remarks to the Author)

Biological macromolecules, including proteins and nucleic acids, have evolved to operate in aqueous environments, and thus their structures and functions are inextricably linked to their interactions with water molecules. Nevertheless, deep and precise biological insight into the energetics and roles of specific water-mediated interactions have remained elusive due to their mobile character and the lack of rigorous approaches to identify their locations.

In this manuscript, Kretsch et al. perform single particle cryogenic electron microscopy analysis of the Tetrahymena ribozyme using a next-generation electron detector, obtaining two independent maps at 2.2-2.3 Å resolution. Even if not by

much, it is the highest resolution cryo-EM structure of an RNA-only sample to date. The authors apply their water-modeling program (SWIM) on this large biomolecule with a significantly solvated core. The quality of the datasets and the ability to cross-reference them with each other and with MD simulations allow the authors to establish an approach that rigorously ascribes features of the density maps to water networks and Mg²⁺ ions with confidence. Their analysis deepens understanding and interpretation of cryoEM maps of RNA and yields fundamental insights into how water supports the folded structure of the macromolecule.

General comments and questions:

1. Could or should the same kind of extensive “consensus/non-consensus” analysis described here be performed on typical cryo-EM maps by using their half maps? Hasn't SWIM already taken data from independent half-maps into account for its placement criteria (Ext. Data Fig. 4)? How is the rest of the paper similar or different from that? For example, in the section starting at 391/Table 2: how do the statistics of the respective half maps of the two independent maps look relative to each other for these metrics? To what extent does the use of a full independent map really facilitate this sort of analysis?
2. If the high stringency of the SWIM modeling was lowered (what would be the right parameters to tweak, and maybe different parameters would give different results), what proportion of the “non-consensus” spots will become consensus? Would that begin to introduce numerous additional false positive/new non-consensus assignments, or does it help to converge the assignments between the two independent maps overall?
3. If SWIM were to be run on the exact same density map with the exact same parameters, would the exact same water/ion placement be obtained each time, or is there some element of randomness/arbitrariness involved in which waters/ions get placed first and which get excluded due to sterics relative to previously placed waters/ions or other factors? If the output isn't guaranteed to be the same each time, could any proportion of the disagreement between “non-consensus” assignments be more attributable to the variation among different instances of running SWIM rather than differences in the two independent maps and their structure/experimental error?
4. Lines 303-305/Fig. 4e: The discussion surrounding this claim seems incomplete relative to the plot shown. If there is statistical difference between consensus and non-consensus waters, the magnitude does not appear to be large, and the larger difference in both statistical power and magnitude is between modeled waters and the MD waters. That larger difference appears to be counter to the general claims associated with Fig. 4. My interpretation of the figure is that the MD-identified waters that were not modeled with SWIM, despite not showing up in the cryoEM data, appear to have significantly decreased mobility in their MD-simulated binding sites.
5. Ext. Data Fig. 2, lines 1060-3: This seems to suggest that the bases matter more than the backbone for assigning water interactions. Is this correct?
6. Fig. 4: Although showing the maximum values in the background of 4c and 4d makes their data look better, it also makes the graph look more cluttered. Since all measurements are bulk, wouldn't more averaging be a better representation of it?
7. Lines 271-272: The distinction between those “two factors” (occupancy and mobility) is not entirely clear to me. Diffuse density must be described by some degree of mobility, but I'm not sure how you really differentiate that into those two discrete factors. Maybe I'm not fully understanding it, but they seem like two sides of the same coin.

Regarding statistics, a couple of small things that could be added:

8. Ext. Data Fig. 1h: Include the R² value for the linear fits somewhere.
9. Ext. Data Fig. 3g: r and p are listed for 3f but not for 3g.

Additional comments:

10. In the introduction the authors have cited several papers that focus on hydration from a structural perspective. It seems relevant to cite Chem Biol. 2004; 11:237-46, as it describes functional approaches to infer whether RNA's 2'-OH makes important functional contributions through interaction with water.
11. I think it is important that the authors state explicitly what their construct is. I believe their previous 3.1A structure had the P1 substrate duplex and omega G present. As I understand it, the construct in the current paper does not. This point also relates to lines 168-171 where the number of waters at the active site are compared to the number of waters in other regions. In this regard, I tend to think of the active site in an RNA as including the bound substrates, since groups on the substrates, i.e. the 2'-OHs of omega G and -1U at the 5'-splice site, contribute substantially to catalysis even though they are not atoms along the reaction coordinate. Thus, it might be premature to conclude that there are similar numbers of waters per nucleotide at the active site as elsewhere. Although beyond the scope of the current work, I'd like to see the comparison made between the active site with docked substrates and other regions of the ribozyme. Along those same lines, a fascinating future study would be to examine how the waters CHANGE between a construct with undocked P1 and one with docked P1 and then to relate this to the energetics of docking.

12. With respect to the detailed discussion of water molecules on p. 10 and Fig. 3, it would be interesting if the authors could review the relevant literature for functional evidence for their significance. In this respect, group I introns have been analyzed by population-based approaches such as nucleotide analog interference mapping, in which the effect of certain functional group substitutions on the group I reaction can be assessed: for example the effect of Rp phosphorothioate substitution (i.e. replacement of a nonbridging oxygen with sulfur, 2'-deoxynucleotide substitution (i.e. replacing 2'-OH with 2'-H) , inosine substitution (replacing the of the exocyclic amine of G with H). It would be interesting if there were effects in these data that could not be explained readily by the RNA structure alone, but now make sense under consideration of water-mediated interactions revealed by the current analysis. One potential caveat is that those assays have tended to start with a 'folded' ground state, which likely could have masked effects from disrupting interactions with water molecules.

13. Lines 121 and 570 (methods)/Ext. Data Fig. 4: Indicates increased the stringency of the SWIM criteria. What are the typical SWIM criteria/what specifically was adjusted?

14. Fig. 3: What is the purpose of having the arrows overlap in the figure?

15. Ext. Data Fig. 5g: Fix number formatting.

16. Line 517: It would be nice to see the molar concentration also listed as mass/volume.

17. Lines 470/473: The "2.2 Å" model is listed twice, presumably one of those sets of pdb identifiers refers to the 2.3 Å models.

18. Lines 562-565: explicitly state here whether the Phenix refinement, SWIM modeling, and ISOLDE refinement were performed on sharpened or unsharpened maps, respectively.

19. The paper uses "Watson-Crick" once and "Watson-Crick-Franklin" once each. Perhaps only use "Watson-Crick-Franklin".

Version 1:

Reviewer comments:

Referee #1

(Remarks to the Author)

The authors have addressed the reviewers' comments and the manuscript has improved. I can recommend publication.

Referee #3

(Remarks to the Author)

In this revised manuscript, assessing the locations of first shell water and Mg²⁺ ions in the Tetrahymena group I intron, the authors have addressed the previous review comments in a thorough and scholarly manner and revised the manuscript appropriately. I believe this manuscript is appropriate for publication and will be enthusiastically received by the journal's readership.

Dear Editor,

Thank you for providing the thoughtful comments from two reviewers. In view of the questions, we re-evaluated our methods of analysis very critically. As a result, we have updated the SWIM algorithm's geometric criteria, such that no Mg^{2+} ion is close to another Mg^{2+} ($<4.5 \text{ \AA}$) and ensuring we are only modeling the first solvation shell (waters and Mg^{2+} must directly bind an RNA atom). With these updates 256 waters and 47 Mg^{2+} ions remain in the 2.2 \AA SWIM model and 281 waters and 47 Mg^{2+} ions remain in the 2.3 \AA SWIM model. There are the same number of consensus waters (134). All changes were manually inspected for chemical plausibility and map quality. These slight changes did not result in any changes in conclusions. The new models have been uploaded for the reviewers and will be uploaded to PDB as a revised model. The following are our responses to each of the questions raised by the reviewers.

Referee #1: cryoEM, molecular dynamics simulations, computational biophysics

This study employed cryo-EM to visualize the Tetrahymena ribozyme at resolutions of 2.2 and 2.3 \AA , uncovering intricate water interactions within the RNA structure. The authors used a Segmentation-guided Water and Ion Modeling (SWIM) to identify water molecules and Mg^{2+} ions from non-proteinaceous electronic density in the cryo-EM maps. This task is challenging due to the inherent ambiguity in interpreting electronic density. The paper is very well written and rigorously performed, synergistically combining molecular dynamics simulations, cryo-EM and computational models for electron density refinement. I support the publication of this article in Nature upon moderate revisions, as outlined below.

Comment #1.1

Through cryo-EM and MD simulations, the authors observed extensive mediation of RNA non canonical interactions by water molecules, with both rigid and flexible waters. Intriguingly, regions where SWIM did not model ordered water showed similar densities in both cryo-EM maps, suggesting these regions should be assigned as water, indicating complex water networks as predicted by MD. While there is general agreement between MD simulations and cryo-EM data, the study notes that the MD-predicted water structure performs poorly in certain regions, particularly within the catalytic core. This suggests limitations in the accuracy of MD simulations in these critical areas, likely due to the constraints of the force-field models. These limitations can be addressed through quantum-classical simulations, which allow for the direct computation of electronic densities in critical regions, including active sites. By comparing and fitting these computed densities with experimental data, as demonstrated by Schwartz et al. (Nat. Commun. 15:3324 (2024)), the accuracy of simulations can be significantly improved.

We agree that more accurate methods, such as quantum-classical simulations as mentioned, may help to overcome limitations of classical force fields. However, those methods are generally intensive and would be necessarily limited to a focused area. The aim of our analysis was to assess simulation accuracy over all regions that were sufficiently resolved experimentally, a region much too large for application of such simulation methods. Further, as we are currently describing the apo-ribozyme, some of the details of the catalytic mechanism (as modeled by QM in Schwartz et al) are less relevant, but certainty would be interesting in a study of the

holo-enzyme. To clarify our choice of method of simulation for the current investigation, we edit the final sentence of the discussion to stress these future developments.

“Second, while the simulation of water dynamics cannot yet accurately model experimental RNA and solvent structural fluctuations fully, the observed agreement warrants further investigation into the utility of MD in interpreting and predicting water structure around RNA, *including various catalytic states of the Tetrahymena ribozyme. In the future, cryo-EM data could be used to evaluate and compare simulation methods, including quantum mechanical simulations and alternative force fields, with particular focus on different parameterizations of water and ions (Schwartz et al. 2024; Li and Merz 2014; Panteva et al. 2015; Grotz et al. 2021).*”

Comment #1.2

The authors report that nMg ions were parameterized using an internally developed parameter file. However, the performance of this Mg model relative to the current leading models in the field remains unclear. Divalent metal ions like Mg²⁺ and Mn²⁺ present unique challenges in simulation modeling. The Li & Merz 12-6-4 models are noteworthy for accurately representing the behavior of these ions, as highlighted in their ability to replicate a wide range of bulk properties (J. Chem. Theory Comput. 10:289–97). When simulating nucleic acid-bound metals, it is essential to include specific pairwise corrections, such as those developed by Panteva and colleagues (J. Phys. Chem. B. 119:15460–70). These corrections are crucial for accurately adjusting the interactions between divalent metal ions (e.g., Mg²⁺, Mn²⁺) and water molecules, which play a significant role in the system investigated here.

There are quite a few proposals in the literature for handling Mg²⁺ interactions with water and RNA. We agree that there has been insufficient assessment and comparison of accuracy between these parametrizations. While that comparison is outside the scope of this paper, the experimental densities and SWIM modeling presented here form one of the targets for the 2024 CASP competition, with a focus on metal ion and water placement (see <https://predictioncenter.org/casp16/target.cgi?id=155&view=all>). This CASP exercise should provide an opportunity to compare the performance of a variety of force field models on this exact system. The results of this experiment will be commented on in a future manuscript but is outside the scope of this current manuscript.

We have updated the discussion to reflect the prospect of using these and similar data to compare simulation methods, including force field parameterization and sampling methods as described in our response to **Comment #1.1**.

Comment #1.3

The authors acknowledge that the ions modeled using SWIM only neutralize about one-quarter of the RNA charge. Do the authors find additional sites for ion binding during MD simulations?

It is unlikely that the RNA is fully neutralized by site-bound ions because a portion of the neutralization is known to originate from diffusely bound ions (Misra and Draper PNAS 2001

<https://doi.org/10.1073/pnas.221234598>). Having said that, we do agree that there are likely other specific ion binding sites with significant occupancy that we have left unmodeled.

Magnesium binding sites were not discussed in detail in the manuscript due to two reasons. First, placement of 20 ions enabled stable simulations without excessive equilibration times, but this initial positioning biases our simulations towards identifying those binding sites. Second, limited simulation time with no enhanced sampling could lead to false negatives, i.e., ion sites not fully sampled in the simulations.

When examining all MD Mg²⁺ ion binding sites, the bias becomes apparent. Out of the 9 MD sites with > 30% occupancy over all simulation time, 7 of the 9 corresponded to a site where Mg²⁺ was placed in the initial conditions in the simulation (**Table R1**, which is part of the new **Supplemental Table 4** in the revised text). For the remaining two sites, a Mg²⁺ was placed nearby. This highlights the bias of the Mg²⁺ sampling towards initial conditions, supporting our caution.

5 of the 9 positions with >30% occupancy in MD were not identified by SWIM in both maps. Many of these sites do have experimental evidence that they are Mg²⁺ ion binding sites, e.g., they are found in previous crystal structures. However, due to the SWIM criteria and diffusivity of density discussed in the text, these sites were either not identified by SWIM or were not placed in a chemically ideal location. For example, the diffuse octahedral shaped density between the backbone of A172 and A173 displayed in **Extended Data Figure 8C**, caused SWIM to place water in that location in one of the maps.

We have now added Supplemental tables with all binding sites in the MD simulations (**Supplemental Table 3** for water, **Supplemental Table 4** for Mg²⁺ ions).

Table R1: Top Mg²⁺ ion binding sites in molecular dynamics simulation, by average occupancy.

RNA atoms bound	Average occupancy	Found in SWIM models?	Placed in the initial position in MD simulation?
A-183:OP A-184:OP A-186:OP	0.99	Yes, both	Yes
A-173:OP	0.96	Yes, one	Yes
A-184:OP A-186:OP	0.94	Yes, both	Yes
U-258:OP	0.85	Yes, both	Yes
G-282:OP	0.72	No	Yes
A-256:OP U-273:OP	0.68	Yes, both	Yes
U-307:OP A-308:OP	0.61	No	Yes
C-211:OP	0.59	No	Nearby - bound to 210:OP
A-97:OP	0.34	No	Nearby - bound to 97:OP and 300:O2'

Referee #3: ribozymes

Biological macromolecules, including proteins and nucleic acids, have evolved to operate in aqueous environments, and thus their structures and functions are inextricably linked to their interactions with water molecules. Nevertheless, deep and precise biological insight into the energetics and roles of specific water-mediated interactions have remained elusive due to their mobile character and the lack of rigorous approaches to identify their locations.

In this manuscript, Kretsch et al. perform single particle cryogenic electron microscopy analysis of the Tetrahymena ribozyme using a next-generation electron detector, obtaining two independent maps at 2.2-2.3 Å resolution. Even if not by much, it is the highest resolution cryo EM structure of an RNA-only sample to date. The authors apply their water-modeling program (SWIM) on this large biomolecule with a significantly solvated core. The quality of the datasets and the ability to cross-reference them with each other and with MD simulations allow the authors to establish an approach that rigorously ascribes features of the density maps to water networks and Mg²⁺ ions with confidence. Their analysis deepens understanding and interpretation of cryoEM maps of RNA and yields fundamental insights into how water supports the folded structure of the macromolecule.

General comments and questions:

1. Could or should the same kind of extensive “consensus/non-consensus” analysis described here be performed on typical cryo-EM maps by using their half maps? Hasn't SWIM already taken data from independent half-maps into account for its placement criteria (Ext. Data Fig. 4)? How is the rest of the paper similar or different from that? For example, in the section starting at 391/Table 2: how do the statistics of the respective half maps of the two independent maps look relative to each other for these metrics? To what extent does the use of a full independent map really facilitate this sort of analysis?

In practice, we believe that SWIM is sufficiently rigorous to be run only on a single map with half-maps. Half-maps are considered in the SWIM procedure, specifically for the Q-score criteria where each molecule placed must have a Q-score above a certain threshold, 0.7 in our manuscript, in the full cryo-EM map and the two half-maps.

In our in depth study shown in this manuscript however, the two independent maps, determined at 2.2 Å and 2.3 Å resolution, were important to investigate the stringency of SWIM and investigate the nature of waters at this resolution. Using two independent maps as opposed to half-maps in our study was advantageous because half-maps can suffer from a reduction in resolvability as they only see half the amount of data. At 2.2 Å, water is just becoming resolvable with a good confidence level, so a reduction of resolvability would critically inhibit our ability to interpret waters with high confidence level. This is evident in the **Table R2** below, which is expanded from **Table 2** from the manuscript, where we can see the two independent full maps agree better in the solvent shell than one full-map and its half maps as shown in bold in this table. We note that their numerical values are different in two half maps probably due to difference in sharpening.

Table R2: Agreement with the immediate solvent shell of the 2.2 Å cryo-EM map where the RNA atoms are well resolved and almost identical between the two independent maps

	Cross-correlation	Mutual information	Area under precision-recall curve	Mathews correlation coefficient
Maximal value	1.00	0.5020	1.00	1.00
Random	0.00	0.0003	0.13	0.03
Independent map	0.90	0.2059	0.75	0.63
Half map A	0.85	0.1817	0.78	0.66
Half map B	0.64	0.1178	0.62	0.53
SWIM model	0.25	0.0019	0.31	0.34
Molecular dynamics	0.23	0.0131	0.31	0.26

This table has also been further improved by limiting volume compared to only around the well-resolved RNA to match analysis elsewhere in the text.

We additionally asked if we could do similar consensus analysis if we ran SWIM on the half-maps alone. We ran SWIM on the half-maps using the same SWIM criteria, except the half-map Q-score check was not conducted. Our new analysis shows that there are 133 and 118 consensus waters for the consensus between the two 2.2 Å half-maps and the two 2.3 Å half-maps respectively. This is in contrast to the 255 and 281 waters identified by SWIM in the 2.2 Å and 2.3 Å map respectively when the half-map are used in the Q-score filter instead. Hence, running our analysis based on consensus of half-maps would have missed a lot of data compared to the analysis we did conduct with the two independent full maps.

2. If the high stringency of the SWIM modeling was lowered (what would be the right parameters to tweak, and maybe different parameters would give different results), what proportion of the “non-consensus” spots will become consensus? Would that begin to introduce numerous additional false positive/new non-consensus assignments, or does it help to converge the assignments between the two independent maps overall?

This is an interesting question. We tested the effects of the SWIM criteria described in Methods by varying the density threshold (originally set at 5) and Q-score threshold (originally set at 0.7 for the peak density of water or ion). We note that changing these values changes the total number of water found in the two maps differently. So, percent consensus is not a good comparison metric as it differs depending on reference, equation A and B below are not equal.

$$(A) \% \text{ consensus } 2.2\text{\AA} \text{ model} = \frac{\# \text{ consensus}}{\# \text{ consensus} + \# 2.2 \text{\AA} \text{ non consensus}}$$

$$(B) \% \text{ consensus } 2.3\text{\AA} \text{ model} = \frac{\# \text{ consensus}}{\# \text{ consensus} + \# 2.3 \text{\AA} \text{ non consensus}}$$

Instead, we use F1-score to compare the results, which account for both false positives and false negatives.

$$(C) F1 = \frac{\# \text{ consensus}}{\# \text{ consensus} + \frac{1}{2}(\# 2.2 \text{\AA} \text{ non consensus} + \# 2.3 \text{\AA} \text{ non consensus})}$$

For the density threshold, we observe an increase in F1 consensus between the two maps as density increases up to roughly 10σ (**Table R3**). This indicates that the higher the density of

the water, the more likely it is to be found in both maps, this conclusion matches conclusions drawn from manuscript **Figure 4A**.

When the water Q-score threshold for full and half-maps is varied, no change in the amount of consensus is observed (**Table R3**). We hypothesize that the lack of pattern with stringency is due to the non-Gaussian nature of these densities as described in the final sections of the manuscript. We note that in manuscript **Figure 2C** we do see that on average the consensus waters have higher Q-score, but this trend is less pronounced than the trend for density in manuscript **Figure 4A** and is not significant for ions.

We have added **Table R3** to Manuscript **Extended Data Figure 4I**.

Table R3: The water assignment and consensus between automatically placed SWIM waters of the two maps. The density threshold above which a water peak is accepted, and the Q-score in the full and half-map threshold above which a water peak is accepted are systematically varied.

Density threshold (σ)	Q_peak_min	Number of waters in 2.2 Å map	Number of waters in 2.3 Å map	Number consensus waters	F1
2	0.7	310	358	148	0.443
3	0.7	310	357	148	0.444
4	0.7	291	326	143	0.464
5	0.7	255	281	134	0.500
6	0.7	210	240	122	0.542
7	0.7	184	203	113	0.584
10	0.7	100	115	70	0.651
15	0.7	26	27	18	0.679
20	0.7	10	5	4	0.533
5	0.5	393	394	214	0.544
5	0.6	363	360	199	0.550
5	0.7	255	281	133	0.500
5	0.8	110	133	56	0.461
5	0.9	11	13	6	0.500

3. If SWIM were to be run on the exact same density map with the exact same parameters, would the exact same water/ion placement be obtained each time, or is there some element of randomness/arbitrariness involved in which waters/ions get placed first and which get excluded due to sterics relative to previously placed waters/ions or other factors? If the output isn't guaranteed to be the same each time, could any proportion of the disagreement between "non consensus" assignments be more attributable to the variation among different instances of running SWIM rather than differences in the two independent maps and their structure/experimental error?

The SWIM method is deterministic and does not involve any randomness, so the output would be the same when applied to the same map and model using the same parameters and version.

4. Lines 303-305/Fig. 4e: The discussion surrounding this claim seems incomplete relative to the plot shown. If there is statistical difference between consensus and non-consensus waters, the

magnitude does not appear to be large, and the larger difference in both statistical power and magnitude is between modeled waters and the MD waters. That larger difference appears to be counter to the general claims associated with Fig. 4. My interpretation of the figure is that the MD-identified waters that were not modeled with SWIM, despite not showing up in the cryoEM data, appear to have significantly decreased mobility in their MD-simulated binding sites.

The noted trend, where we see MD-waters that are not SWIM-identified waters as having low RMSF, was due to a confusing aspect of our analysis, which we now correct.

In the original analysis, an MD water binding site was simply defined by the RNA atoms and the water coordinates. This was an acceptable definition when comparing binding sites globally, however, when analysing the motion in the binding site itself, we now believe it was insufficient and also not the appropriate comparison to cryo-EM data.

In particular, when we look at only high occupancy water (waters present >50% of the time), we see bimodal RMSF values (**Figure R1**), and this trend is also visible in the original **Figure 4E**. The first peak is clearly centered around ~ 0.5 Å, a value expected of a single well-defined water binding site. However, the second peak is at ~ 2.5 Å. The binding sites in this second peak are defined as the same binding site by our analysis, e.g. binds same RNA atoms, however, it is likely they represent two chemical distinct binding sites water is able to bind in site A and B, both binding to same RNA atoms. This dual-water binding site is much more common amongst the cryo-EM binding sites, likely because they are expected to be in highly ordered regions. So we now believe the RMSF values were obscured by dual-binding sites, and may not reflect local deviations accurately.

Figure R1: For each water binding site with >50% occupancy across all simulations, the RMSF of each water while it remained in the binding pocket, defined by RNA atoms bound only, is plotted. Binding pocket in this plot is defined by RNA atoms alone, now the manuscript defines MD binding pocket by RNA atoms bound and position.

Hence prompted by this review comment, we decided to redefine MD binding sites to be more directly comparable with how we defined the cryo-EM sites – we included a positional requirement in addition to the RNA atoms bound. Briefly, we now prepare maps of the water density from the MD simulations – closely analogous to the cryo-EM maps – and we identify all peaks in the MD density of waters, sodium ions, and magnesium ions in the simulation. These peak locations are then considered the set of binding sites. Any MD water that is within 2 Å of

the peak and binds the same RNA atoms as another MD water is said to occupy the same binding site. Likewise, we can label each of these locations as overlapping or not with any SWIM cryo-EM waters. With this more direct comparison, we have remade **Figure 4**, as well as all Extended Data figures that depended on our definition of MD binding site. As can be seen in **Figure R2** our conclusions remain unchanged, but are now more clearly supported by the analysis. In particular, it is clear that a stronger peak density at each site accounts for whether it will be selected by SWIM in both maps, one map, or none (**Fig. R2A**). We then explain the biophysical properties of the water that may be causing a difference in peak density (**Fig R2B**). These biophysical properties cannot be measured in the cryo-EM maps at this resolution, but we show the MD simulations water density is well correlated with the cryo-EM density, supporting our use of the simulations in interpreting these biophysical properties (**Fig. R2C-D**). The MD analysis suggests that either low occupancy or high mobility (RMSF) can account for different low-peak density sites that are challenging for SWIM (**Fig. R2E-F**), with positional spread within the site (RMSF) being the major factor.

Figure R2: Updated version of manuscript **Figure 4**. With the updated definition of MD binding site, and panels clearly showing that peak density is a major difference between consensus and non-consensuses, and explain the biophysical reasons for peak density difference.

5. Ext. Data Fig. 2, lines 1060-3: This seems to suggest that the bases matter more than the backbone for assigning water interactions. Is this correct?

The data displayed in **Extended Data Figure 2** show that the atoms in the bases are better resolved than those in the backbone. Interestingly, however, as seen in **Extended Data Figure 7C**, SWIM assigns a similar number of cryo-EM density peaks near backbone atoms than near base atoms. Additionally we see no difference in Q-score between waters bound to bases and those bound to backbone (**Figure R3**).

Instead, water Q-score seems primarily explained by the number of interacting RNA atoms; the more RNA atoms a water interacts with the higher Q-score it has in general (**Figure R3**), likely due to increased order of that water. This observation has been added to manuscript **Extended Data Figure 4L-M**.

We have not commented generally on whether base-water or backbone-water interactions are more important for stability because we are at a resolution where water is just becoming resolvable, with density potentially biasing which sites SWIM can assign water atoms. It will be fascinating to explore further determinants of water visibility in cryo-EM maps in the future, especially if resolution and signal-to-noise ratios can be further increased.

Figure R3: SWIM assigned waters and magnesium ions from the 2.2 and 2.3 Å map are categorized by the type of atoms they are bound to (left, both maps) and the number of RNA atoms they interact with (right, separated by map). The Q-score is plotted with the median and quartiles plotted as horizontal lines. The number of waters or magnesium ions is labeled above the plot. While the trend for the 2.2 Å model shows an increase in Q-score with more interacting atoms, this trend is weaker for the 2.3 Å model, and neither are statistically significant (linear regression, $p > 0.5$).

6. Fig. 4: Although showing the maximum values in the background of 4c and 4d makes their data look better, it also makes the graph look more cluttered. Since all measurements are bulk, wouldn't more averaging be a better representation of it?

Yes, this is a fair point. In particular for the plots shown here for water where we assume sampling is quite good in each simulation, we agree it is ok to base conclusions from the bulk measurements. We have removed the maximum values from the plots. See revised version of **Figure 4** above (**Figure R2**).

7. Lines 271-272: The distinction between those “two factors” (occupancy and mobility) is not entirely clear to me. Diffuse density must be described by some degree of mobility, but I’m not sure how you really differentiate that into those two discrete factors. Maybe I’m not fully understanding it, but they seem like two sides of the same coin.

Yes, these terms are indeed confusing. We were observing the peak density at the binding site area. Looking at density at a *peak*, both experimentally and in simulation, we are considering how often a water is precisely at that peak coordinate. The peak density can be reduced in two ways. First, for occupancy, water is not always present anywhere in the site, which reduces the mean density and peak density. Second, for positional spread, waters are present in the site but they move around and hence are not found precisely at the peak coordinate, diffusing out the density locally.

To clarify we have modified the text and updated manuscript above (**Figure R2**) which now explicitly labels the density as “peak density”. We have additionally added examples of high and low occupancy and RMSF sites in **Figure 4B** (**Figure R2** above).

“Peak density can be reduced by two factors: occupancy, (i.e., how often water is present *anywhere in the site*), and high positional spread (i.e. water diffuses ~~in and out of~~ within the binding sites *and is not always localized to the peak coordinate*).”

Regarding statistics, a couple of small things that could be added:

8. Ext. Data Fig. 1h: Include the R2 value for the linear fits somewhere.

The plots in manuscript **Extended Data Figure 1H-I** have been updated as seen below in **Figure R4**, quoting the R² for the linear regression.

Figure R4: Plots of particle number against the reciprocal squared resolution for (left) 2.2 Å and (right) 2.3 Å maps. The B-factor was calculated as twice the linearly fitted slope and the R² value is reported.

9. Ext. Data Fig. 3g: r and p are listed for 3f but not for 3g.

Since RMSF and B-factor are monotonically related, r and p are the same. We have added these r and p values to the plot for clarity in manuscript **Extended Data Figure 3**.

Additional comments:

10. In the introduction the authors have cited several papers that focus on hydration from a structural perspective. It seems relevant to cite Chem Biol. 2004; 11:237-46, as it describes functional approaches to infer whether RNA's 2'-OH makes important functional contributions through interaction with water.

Thank-you for the recommendation. This reference has been added to support the following statement in the introduction.

“Water has been implicated in the stability, catalysis, and dynamics of RNA both independently and in collaboration with ions.”

11. I think it is important that the authors state explicitly what their construct is. I believe their previous 3.1A structure had the P1 substrate duplex and omega G present. As I understand it, the construct in the current paper does not. This point also relates to lines 168-171 where the number of waters at the active site are compared to the number of waters in other regions. In this regard, I tend to think of the active site in an RNA as including the bound substrates, since groups on the substrates, i.e. the 2'-OHs of omega G and -1U at the 5'-splice site, contribute substantially to catalysis even though they are not atoms along the reaction coordinate. Thus, it might be premature to conclude that there are similar numbers of waters per nucleotide at the active site as elsewhere. Although beyond the scope of the current work, I'd like to see the comparison made between the active site with docked substrates and other regions of the ribozyme. Along those same lines, a fascinating future study would be to examine how the waters CHANGE between a construct with undocked P1 and one with docked P1 and then to relate this to the energetics of docking.

We appreciate the reviewer's careful observations regarding the construct. Indeed, the construct used in the current study is the L-21 ScaI ribozyme, a linear form of the self-splicing intron, spanning nucleotides 22–409. This construct does not include the P1 substrate duplex or ω G. We now clarify this in the text and Methods section to provide precise context for readers.

In our previous 2021 study (Su et al Nature <https://doi.org/10.1038/s41586-021-03803-w>), we employed the L-16 ScaI ribozyme construct with an extended IGS (nts 17–27), which bound two oligonucleotide substrates (S1 and S2). This construct formed a 4-bp P10 and a 6-bp P1 with S1 and S2. The structural comparison between L-16 and L-21 constructs reveals that both share a similar overall architecture, including flexible regions and active site, with one key difference: the IGS in the L-16 undergoes a significant conformational change to form P1. We have added this structural comparison for the reviewer (**Figure R5** below), illustrating the impact of substrate binding on ribozyme conformation.

Regarding the reviewer’s question about water molecules at the active site, we agree that it may be premature to conclude that the active site has a comparable density of waters per nucleotide as other regions, especially considering the catalytic contributions from substrate groups (e.g., 2’-OHs of omega G and -1U at the 5’-splice site). We have revised the text to reflect this caution, recognizing that the functional composition of the active site may involve these specific substrate groups.

“Interestingly, the catalytic active site of *Tetrahymena* ribozyme had a similar number of waters per nucleotide as the other regions of the ribozyme (Table 1), indicating that ordered waters may be important for tertiary interactions generally and *there is not a particularly elaborate water structure in the active site when substrate groups are absent* not just for the enzymatic activity of RNA.”

Finally, we acknowledge the reviewer’s suggestion for a comparative study examining water molecule differences between undocked and docked P1 states and relating this to docking energetics. This is indeed a fascinating direction for future work, and we look forward to exploring how hydration and water dynamics influence docking energetics in these ribozyme constructs.

Figure R5: Conformational changes between apo and holo states of the *Tetrahymena* ribozyme.

(a) Superposition of the apo and holo states reveals major structural differences in the IGS. (b) The root-mean-square deviation (RMSD) map shows regions of structural changes (color-coded from blue to red).

12. With respect to the detailed discussion of water molecules on p. 10 and Fig. 3, it would be interesting if the authors could review the relevant literature for functional evidence for their significance. In this respect, group I introns have been analyzed by population-based approaches such as nucleotide analog interference mapping, in which the effect of certain functional group substitutions on the group I reaction can be assessed: for example the effect of Rp phosphorothioate substitution (i.e. replacement of a nonbridging oxygen with sulfur, 2’-deoxynucleotide substitution (i.e. replacing 2’-OH with 2’-H) , inosine substitution (replacing the of the exocyclic amine of G with H). It would be interesting if there were effects in these

data that could not be explained readily by the RNA structure alone, but now make sense under consideration of water-mediated interactions revealed by the current analysis. One potential caveat is that those assays have tended to start with a ‘folded’ ground state, which likely could have masked effects from disrupting interactions with water molecules.

Nucleotide analog interference mapping (NAIM) is a rich source of functional data, and we appreciate the prompt to review these papers. Analysing NAIM-identified atoms and interacting partners, we see that most of these atoms and interactions involve direct RNA-RNA hydrogen bonds (**Table R4**, Szewczak et al 1998, Strobel et al 1998, Strauss and Strobel 2002). In particular, the majority of NAIM-identified atoms (42 of 50) that show high interference are also involved in direct hydrogen bonding to other RNA atoms. Out of the remaining 8 atoms, there are two whose functional importance might be explained by new features in our maps and SWIM analysis: A256 N6 and A308 N7, which are involved in a water-mediated interaction. For A114 O2’ and six N7 atoms, our SWIM models do not reveal an interaction, although a water- or ion-mediated interaction is geometrically possible and some cases have cryo-EM density that would support that interaction, as documented in **Table R4**.

We do note that there is not a perfect concordance of NAIM with SWIM. In particular, when we examine the 20 adenines and guanines that form water-mediated interactions in our cryo-EM models, only 3 of 20 were identified in NAIM and two of these three are also involved in direct hydrogen bonds which presumably explain the NAIM signal (**Table R5**). In general, it appears that NAIM is less sensitive to water-mediated interactions, perhaps because these interactions can accommodate and adapt to atom substitution more readily than a direct hydrogen bond or because these interactions are less important for stability or catalysis. Detailed dissection of the observed water networks through functional experiments will be interesting future work; to help prompt such studies, we have added **Table R4** to the paper as **Supplemental Table S2**, and added a clause to the following sentence in the discussion:

“These ordered waters could be important for the pre-structuring of catalytically important ions or to reduce the entropic penalty of desolvation upon substrate binding *and may be may be important for understanding currently unexplained effects from nucleotide analog interference mapping experiments (Supplemental Table 2).*”

Table R4: Atoms identified by NAIM to have strong interference of catalytic activity, as labeled by refs, and the interaction observed in the structures presented. (**Supplemental Table 2** in manuscript)

Interacting atoms ^{1,2}	Reported in	Interaction in structure
A256 N6	Ortoleva-Donnelly et al RNA 1998	Water-mediated interaction U300 OP
A308 N7	Ortoleva-Donnelly et al RNA 1998	Water-mediated interaction A306 N3
A308 N6	Ortoleva-Donnelly et al RNA 1998	Direct H-bond U267 O4 Water-mediated interaction A306 N3
A114 O2’	Ortoleva-Donnelly et al RNA 1998	No interaction Possible water-mediated interaction with A207 N1, weak density.

A114 N7	Ortoleva-Donnelly et al RNA 1998	No interaction In plane with A106 (H-bond of A114 N6 and A206 N3) could have water/ion mediated interaction, but not clear density
A207 N7	Ortoleva-Donnelly et al RNA 1998	No interaction In-plane with A113 (H-bond of A113 N6 and A207 N3) could have water/ion mediated interaction, but not clear density.
A256 N7	Ortoleva-Donnelly et al RNA 1998	No interaction Density supports a water bound to this atom which would coordinate the Mg ²⁺ ion which coordinates the OP of A256, U273.
A261 N7	Ortoleva-Donnelly et al RNA 1998	No interaction Density supports ion-mediated interaction with A265 OP, modeled as Mg ²⁺ ion, could be a monovalent ion.
A306 N7	Ortoleva-Donnelly et al RNA 1998	No interaction Density supports water/ion mediated interaction with A261 OP.
A95 HN6	Ortoleva-Donnelly et al RNA 1998	Direct H-bond U56 O2'
A95 N7	Ortoleva-Donnelly et al RNA 1998	Direct H-bond U56 HO2'
A97 O2'	Ortoleva-Donnelly et al RNA 1998	Direct H-bond U59 O2' G92 N2
G111 N2	Strobel and Shetty 1997, Ortoleva-Donnelly et al Biochemistry 1998	Direct H-bond C208 O2
G112 N2	Strobel and Shetty 1997, Ortoleva-Donnelly et al Biochemistry 1998	Direct H-bond C209 O2
A210 N6	Strobel et al 1998, Ortoleva-Donnelly et al RNA 1998	Direct H-bond A46 N3
A210 O2'	Ortoleva-Donnelly et al RNA 1998	Direct H-bond C211 O2'
G212 N2	Szewczak et al 1998	Direct H-bond C109 O2 A184 N3
A218 N6	Ortoleva-Donnelly et al RNA 1998	Direct H-bond U273 O2'
A218 O2'	Ortoleva-Donnelly et al RNA 1998	Direct H-bond C102 O2'
A219 N6	Ortoleva-Donnelly et al RNA 1998	Direct H-bond G254 O2' and N3
A256 HO2'	Ortoleva-Donnelly et al RNA 1998	Direct H-bond G272 O2'
A261 O2'	Szewczak et al 1998, Ortoleva-Donnelly et al RNA 1998	Direct H-bond G264 OP
A270 N7	Ortoleva-Donnelly et al RNA 1998	Direct H-bond A103 N6

A270 O2'		Ortoleva-Donnelly et al RNA 1998	Direct H-bond G272 N7
G303 N2		Strobel and Shetty 1997, Ortoleva-Donnelly et al Biochemistry 1998, Szewczak et al 1998	Direct H-bond A302 OP G303 OP
A306 O2'		Ortoleva-Donnelly et al RNA 1998	Direct H-bond A261 N3
A97 N7	U300 H3	Szewczak et al 1998, Ortoleva-Donnelly et al RNA 1998	Direct H-bond
A97 HN6	U300 O4	Szewczak et al 1998, Ortoleva-Donnelly et al RNA 1998	Direct H-bond
A114 HN6	A206 N3	Strobel et al 1998, Ortoleva-Donnelly et al RNA 1998	Direct H-bond
A114 HN6	A206 O2'	Strobel et al 1998, Ortoleva-Donnelly et al RNA 1998	Direct H-bond
G150 HO2'	A152 N7	Strauss-Soukup and Strobel 2000	Direct H-bond
G150 HN2	A153 OP	Strauss-Soukup and Strobel 2000	Direct H-bond
G150 HN2	A153 N7	Strauss-Soukup and Strobel 2000	Direct H-bond
A151 N1	A248 HN6	Strauss-Soukup and Strobel 2000	Direct H-bond
A151 HN6	A248 N1	Strauss-Soukup and Strobel 2000	Direct H-bond
A152 HO2'	U224 HO2'	Strauss-Soukup and Strobel 2000	Direct H-bond
A152 N3	U224 HO2'	Strauss-Soukup and Strobel 2000	Direct H-bond
A152 HN6	G250 O2'	Strauss-Soukup and Strobel 2000	Direct H-bond
A153 O2'	C223 HO2'	Strauss-Soukup and Strobel 2000	Direct H-bond
A153 HO2'	C223 O2	Strauss-Soukup and Strobel 2000	Direct H-bond
A153 N1	G250 HO2'	Strauss-Soukup and Strobel 2000	Direct H-bond
A153 N3	G250 HN2	Strauss-Soukup and Strobel 2000	Direct H-bond
A207 HN6	A213 N3	Strobel et al 1998, Ortoleva-Donnelly et al RNA 1998	Direct H-bond
A207 HN6	A213 O2'	Strobel et al 1998, Ortoleva-Donnelly et al RNA 1998	Direct H-bond
C209 HN4	U305 O4	Szewczak et al 1998	Direct H-bond
C223 O2	G250 HN2	Strauss-Soukup and Strobel	Direct H-bond

		2000	
C223 N3	G250 N1	Strauss-Soukup and Strobel 2000	Direct H-bond
C223 HN4	G250 O6	Strauss-Soukup and Strobel 2000	Direct H-bond
U224 N3	A238 N7	Strauss-Soukup and Strobel 2000	Direct H-bond
U224 O2	A248 HN6	Strauss-Soukup and Strobel 2000	Direct H-bond

¹ When only one atom is listed, the literature identified this atom as interfering with catalysis when substituted, however, did not identify what atom it was interacting with.

² The following interaction were not included in the table as they are not applicable in apo state studied here:

- J4/5 - P1 helix Szewczak et al 1998, Strobel et al 1998, Strauss-Soukup and Strobel 2000
- J8/7 - P1 helix Szewczak et al 1998, Strauss-Soukup and Strobel 2000
- G22-G27 Strobel and Shetty 1997, Ortoleva-Donnelly et al Biochemistry 1998
- 300-302 interactions Szewczak et al 1998, Ortoleva-Donnelly et al RNA 1998

Table R5: The interference of atoms identified as being involved in water-mediated interactions in the structures presented.

Atom ¹	Water-mediated interacting atoms	Direct H-bond atoms	Nucleotide analogue interference ²
A-304:N6	A-269:N3, A-270:O4'		N
A-218:N6	G-254:O2', C-255:O4', U-273:O3'	U273 O2'	Y
G-215:N2	A-105:O2', U-258:O2, U-258:O2'		N
G-272:N2	A-218:N1, C-255:N3	C102 O2	N
A-308:N6	A-306:N3	U267 O4	Y
A-104:N7, A-105:N6, A-269:N6	A-103:N3		N
G-309:N2	A-342:N3, A-342:O2'	C266 O2	N
A-171:N7	U-168:OP, C-170:OP		N
A-183:N6	U-168:OP		N
A-173:N6	U-167:N3, A-171:N3, A-171:O2', G-174:O6	U167 O4	N
A-256:N6	U-300:OP		Y
G-110:N2	U-168:O2, A-183:N1	C211 O2	N
A-270:N6	G-257:O3'	A103 N1	N
A-184:N6	G-212:O2'		N
A-183:N6	U-168:O4', G-212:O2'		N
G-92:N2	C-98:OP	U59 O2	N
G-220:N2	U-253:O2'	U253 O2	N
A-153:N6	G-150:O4'		N
A-151:N7	G-150:O2'		N
G-163:N2	A-139:OP, A-140:OP, G-141:O6,	A-140:N7	N

¹ Consensus waters that bound N6 or N7 of adenines or N2 of guanines were selected for comparison to NAIM results. Waters were not considered if they only bound one nucleotide or they only bound nucleotides along a standard A-form helix. Water mediated interactions were defined as atoms that bound the same water molecule.
² For adenines:(Ortoleva-Donnelly et al RNA 1998) For guanines: (Ortoleva-Donnelly et al Biochemistry 1998)

13. Lines 121 and 570 (methods)/Ext. Data Fig. 4: Indicates increased the stringency of the SWIM criteria. What are the typical SWIM criteria/what specifically was adjusted?

In the previous report of SWIM (Zhang et al. Cell 2020 <https://doi.org/10.1038/s41422-020-00432-2>), only a density threshold was used, set at 2σ . This was increased to 5σ for this study, to account for the increased level of noise we would expect in our 2.2 and 2.3 Å maps compared to the previous 1.34 Å map of apoferritin. Further, we added two more stringency criteria: Q_peak_min (0.7) and Q_res_min (0.6), which test Q-scores of the peak position and residues of nearby atoms respectively. Q_peak_min is additionally tested in both half maps. These new criteria should lead to placing waters and ions in density with more uniform Gaussian peaks. Using Q-scores as a criteria may place water and ions in regions of higher confidence but may also avoid placing waters and ions on areas that are more diffuse and hence not well defined as a uniform Gaussian peak.

A sentence has been modified in the main text to clarify that new metrics were introduced:

“Several SWIM criteria were updated *or introduced* to be more stringent and reduce the likelihood of modeling water in noise peaks at the 2.2-2.3 Å resolution observed here, resulting in conservative peak identification (Extended Data Fig. 4A-H, Online Methods).”

A sentence has been added to the methods to explain previous criteria used:

“Previously, SWIM was run on a 1.34 Å map of apoferritin, using a 2σ density threshold and with no Q-score or half-map threshold (Zhang et al. 2020). In this study, the density threshold has been increased to 5σ and minimum Q-score criteria were imposed, including Q-scores in full and half-maps and Q-scores of bound nucleotides in the full maps for increased stringency.”

14. Fig. 3: What is the purpose of having the arrows overlap in the figure?

The arrows point to the location of the focused area. We have changed the ordering of the panel to prioritize clarity of the arrows, please see **Figure R6** below. We additionally identified an incorrect labeling in manuscript **Figure 3I**. The interacting residue should be A306; this has been corrected, and text modified accordingly.

Figure R6. Updated ordering of panels of manuscript **Figure 3** to reduce arrow overlap.

15. Ext. Data Fig. 5g: Fix number formatting.

The text size has been adjusted to fit, see **Figure R7** below.

Figure R7: updates number size of manuscript **Extended Data Figure 5G**.

16. Line 517: It would be nice to see the molar concentration also listed as mass/volume.

This has been added to the cryo-EM methods section.

17. Lines 470/473: The “2.2 Å” model is listed twice, presumably one of those sets of pdb identifiers refers to the 2.3 Å models.

Thank you for catching this oversight; this has been corrected.

18. Lines 562-565: explicitly state here whether the Phenix refinement, SWIM modeling, and ISOLDE refinement were performed on sharpened or unsharpened maps, respectively.

Maps were sharpened using Phenix.auto_sharpen, using resolution parameters of 2.2 and 2.3 for the 2.2 and 2.3Å maps respectively. Models were refined using ISOLDE into the sharpened full maps. SWIM modeling was performed using sharpened full and sharpened half maps. This has been clarified in the methods.

19. The paper uses “Watson-Crick” once and “Watson-Crick-Franklin” once each. Perhaps only use “Watson-Crick-Franklin”.

All uses have been updated to Watson-Crick-Franklin.